# Alberta Wells Dataset: Pinpointing Oil and Gas Wells from Satellite Imagery

## Abstract

Millions of abandoned oil and gas wells are scattered across the world, leaching methane into the atmosphere and toxic compounds into the groundwater. Many of these locations are unknown, preventing the wells from being plugged and their polluting effects averted. Remote sensing is a relatively unexplored tool for pinpointing abandoned wells at scale. We introduce the first large-scale benchmark dataset[1] for this problem, leveraging high-resolution multi-spectral satellite imagery from Planet Labs. Our curated dataset comprises over 213,000 wells (abandoned, suspended, and active) from Alberta, a region with especially high well density, sourced from the Alberta Energy Regulator and verified by domain experts. We evaluate baseline algorithms for well detection and segmentation, showing the promise of computer vision approaches but also significant room for improvement.

## 1 Introduction

Across the world, there are millions of abandoned oil and gas wells left to degrade by the companies or individuals that built them. No longer producing usable fossil fuels, these wells nonetheless have a significant impact on the environment, with many of them leaking significant quantities of methane, a powerful greenhouse gas, into the atmosphere. In Canada, an estimated 370,000 abandoned wells produce methane equivalent to half a million metric tons of $CO_2$ annually (Williams et al., 2020; ECCC, 2024), while in the U.S. there are an estimated 4 million abandoned wells (Williams et al., 2020), releasing over five million metric tons of $CO_2$ equivalent emissions per year. Abandoned wells also pose health and safety concerns, in particular by leaching toxic chemicals into the groundwater of surrounding communities (Cahill et al., 2019).

It is possible to plug abandoned wells to mitigate the harms associated with them (with so-called "super-emitter" wells an especially high priority (Riddick et al., 2024; Kang et al., 2016)). However, a significant fraction of abandoned wells remain unknown. In Pennsylvania, as much as 90% of abandoned wells are estimated to be unrecorded (Kang et al., 2016). In Canada, abandoned wells have been described as the most uncertain source of methane emissions nationally due to the poor quality of data surrounding them (Williams et al., 2020).

With the advent of large-scale remote sensing datasets and powerful machine learning tools to process them, it has become possible to label and monitor the built environment as never before (Rolf et al., 2024). Many such works have focused on opportunities to use remote sensing to accelerate climate action and environmental protection, and oil and gas infrastructure has increasingly been an object of scrutiny (see e.g. (Yang et al., 2013; Sheng et al., 2020)).

In this paper, we present the first large-scale machine-learning dataset for pinpointing onshore oil and gas wells, encompassing abandoned, suspended, and active wells. Our main contributions are as follows:

- We introduce the Alberta Wells Dataset, which includes information on over 200k abandoned, suspended, and active onshore wells with high-resolution satellite imagery.
- We frame the problem of identification of wells as a challenge for object detection and binary segmentation.

---

[1] Dataset available at: `https://figshare.com/s/bdb097730714ee82fcb0`

- We evaluate a wide range of deep learning algorithms commonly used for similar tasks, finding promising performance but opportunities for significant improvement.

We hope that this work will represent a step towards scalable identification of abandoned well sites and the reduction of their deleterious effects on our climate and environment.

## 2 PREVIOUS WORK

Hundreds of satellites continuously monitor the Earth's surface, generating petabyte-scale remote sensing datasets (Rolf et al., 2024). With advancements in hardware, the quality of remote sensing images has significantly improved in terms of spatial and temporal resolution. High-quality remote sensing data are available through state-funded projects like Sentinel and Landsat, and more recently through private enterprises such as Planet Labs (PBC, 2024). Increasingly, machine learning has been used to parse such raw data, including in a wide range of applications for tackling climate change (Yang et al., 2013). Benchmark datasets in this area have included tasks in land use and land cover (LULC) estimation (Sumbul et al., 2019), crop classification (Sykas et al., 2022; Tseng et al., 2021), species distribution modeling (Teng et al., 2023), and forest monitoring (Ioannis Bountos et al., 2023).Some datasets like SpaceNet 7 Etten et al. (2021) include a few cases of study sites with oil wells, although the dataset was developed for multi-temporal urban monitoring.

Within this area of research, an increasing body of work has considered the problem of detecting artifacts associated with oil and gas operations. The detection of oil spills using a combination of remote sensing and machine learning has been widely explored (Chen et al., 2017; Wang et al., 2023a; Yang et al., 2022). Recently, the detection of oil and gas infrastructure has also been investigated (Sheng et al., 2020; Prajapati et al., 2022), with some studies focusing on the goal of estimating methane emissions (Zhu et al., 2022; Omara et al., 2023). The dataset by (Sheng et al., 2020) includes 7,066 aerial images, with 149 images of oil refineries. The METER-ML dataset (Zhu et al., 2022) comprises 86,599 georeferenced images in the U.S. labeled for methane sources. The OGIM v1 dataset (Omara et al., 2023) includes 2.6 million point locations of major facilities. A dataset by (Chang et al., 2023) features 1,388 images of pipelines in the Arctic, while a dataset by (Wang et al., 2023b) includes 3,266 images of heavy-polluting enterprises with 0.25 m resolution.

The problem of detection of oil and gas wells has also been proposed by a number of authors. Existing datasets, however, are quite small (500-10,000 samples) and typically are limited to a small region and contain only active wells, limiting their applicability in the context of identifying abandoned or suspended wells as summarized in Table 1. Most of these studies have primarily focused on basic machine learning algorithms for well detection due to the limited sample size.

Table 1: Previous datasets in which remote sensing algorithms are applied to detect oil and gas wells. "N/A" is given for datasets which do not indicate the number of individual wells in the dataset.

| Dataset | O&G Well Count | Total Well Images | Resolution (m/px) | Geography | Imagery Source |
|---|---|---|---|---|---|
| NEPU-OWOD V1.0 (Wang et al., 2021) | 1,192 | 432 | 0.41 | Daqing City, China | |
| NEPU-OWOD V3.0 (Zhang et al., 2023) | 3,749 | 722 | 0.48 | China & California | |
| Oil Well Dataset (Shi et al., 2021) | N/A | 5,895 | 0.26 | Daqing City, China | Google Earth |
| O&G Infrastructure (Guisiano et al., 2024) | 630 | 930 | 0.15 - 1 | Permian Basin, USA | |
| Well Pad Dataset (Ramachandran et al., 2024) | 12,490 | 10,432 | 0.3-0.7 | Permian and Denver Basins, USA | |
| NEPU-OWS V1.0 (Wu et al., 2023b) | N/A | 1,200 | 10 | Russia | Sentinel-2 |
| NEPU-OWSV2.0 (Wu et al., 2023a) | N/A | 120 | 10/20/60 | Austin, USA | |
| **Alberta Well Dataset (Ours)** | **213,447** | **94,343** | 3 | Alberta, Canada | Planet Labs |

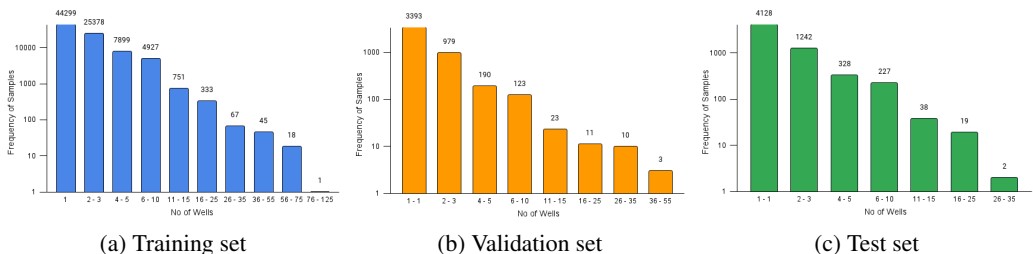

(a) Training set       (b) Validation set       (c) Test set

Figure 1: Distribution of the number of individual wells in positive samples from the dataset. We also include an equal number of images with no wells at all.

## 3 ALBERTA WELLS DATASET

In this paper, we introduce the **Alberta Wells Dataset** for oil and gas well detection. The dataset is drawn from the province of Alberta, Canada, a region with the third-largest oil reserves in the world and a substantial number of oil and gas wells, many of which have been present for over a century. The entire province of Alberta (an area larger than the UK and Germany combined) encompasses a diverse range of geographical zones and is highly diverse for a landlocked region, including prairies, lakes, forests, and mountains. The dataset contains over 94,000 patches of satellite imagery acquired from Planet Labs (PBC, 2024), covering more than 213,000 individual wells. Each patch is annotated with labels for both segmentation and bounding box localization. The annotations are based on data from the Alberta Energy Regulator, quality-controlled by domain experts.

Our dataset attempts to maximize the amount of data available for learning by including a mixture of active and suspended wells alongside abandoned wells. These types of wells appear overall similar in satellite imagery. In contrast to abandoned wells, "suspended" refers to wells that have merely paused operations temporarily, though this designation can be inaccurate, and some wells are classified as suspended for long enough that they are truly abandoned. Active wells are those that are currently in operation.

To simulate real-world conditions, we ensure a varied density of wells per image, as highlighted in Figure 1. We also include satellite imagery patches with no wells present from areas nearby to areas with wells, ensuring no overlap between the samples. This balanced dataset maintains an equal distribution of well and non-well images. Table 2 details the total sample count in each dataset split, alongside the number of well and non-well patches.

### 3.1 WELL DATA COLLECTION, QUALITY CONTROL & PATCH CREATION

The Alberta Energy Regulator (AER) oversees the energy industry in the province, ensuring companies adhere to regulations as they develop oil and gas resources. AER publishes AER ST37 (AER, 2024), a monthly list of all wells reported in Alberta, detailing their geographic location, mode of operation, license status, and type of product being extracted, among other attributes. This data provides a metadata (.txt) file and a .shp shape file, where each entry represents a unique geo-location point per site but often contains duplicates. However, this data cannot be used directly because the license status or mode of operation does not always correlate with the actual status of the well and often contains duplicates. Therefore, we work with domain experts to perform quality control on the dataset as illustrated in Figure 2.

Table 2: Statistics of instances and wells represented across the Alberta Wells Dataset.

| Split | Count Patches Total | Count Wells Patches | Count Non-Wells Patches | Count of Well Type in Wells Patches of Split | | |
|---|---|---|---|---|---|---|
| | | | | Abandoned | Suspended | Active |
| Train | 167436 | 83718 | 83718 | 46342 | 47595 | 100294 |
| Validation | 9463 | 4731 | 4731 | 3166 | 2671 | 2406 |
| Test | 11789 | 5894 | 5894 | 4024 | 3609 | 3340 |

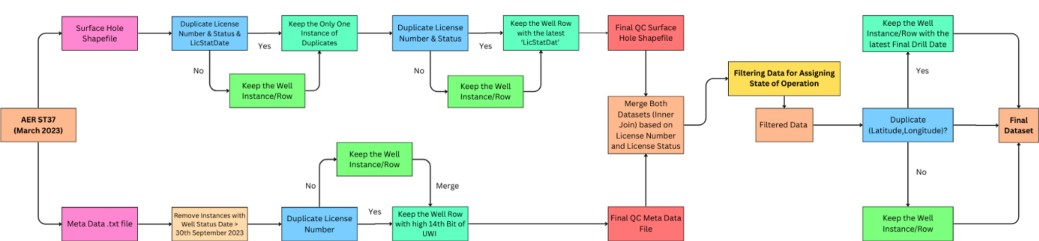

Figure 2: AER ST37 Dataset Cleaning and Quality Control

First, we remove duplicate entries from the well metadata, which often contain multiple instances of the same well identified by duplicate license numbers. We resolve these duplicates by retaining the most recent update. A similar approach is applied to the shapefile, where duplicates are resolved using the license date. Afterward, we merge both datasets and filter the data, categorizing the wells as active, abandoned, or suspended based on specific criteria developed in consultation with domain experts, as shown in Table 3. We check for duplicate location coordinates in the dataset and resolve them by retaining the instance with the latest drill date. Finally, we ensure all the well instances in the dataset are indeed within the boundaries of Alberta. The raw metadata file has around 637,000 instances, while the surface hole geometry file has around 512,000 instances. After quality control and filtering, we have around 217,000 instances.

After filtering and performing quality control on the datasets with domain experts, we calculate the geographical bounds covered by the well instances across the province and divide the region into non overlapping square image patches, each covering an area of 1.1025 sq km (with sides of 1050m). These images include various numbers of individual wells (see Fig. 1), and we ensure that an approximately equal number of patches exist with and without wells. As a result of this process, some samples were excluded due to being located outside Alberta's geographical boundaries, leading to a final total of approximately 213,000 well instances in the dataset patches.

## 3.2 DATASET SPLITTING

To create a well-distributed dataset that represents various geographical regions and offers a diverse dataset for evaluation and testing, we developed a splitting algorithm (see Algorithm 1). Our splitting approach focuses on balancing regions, not individual examples, ensuring that both the training and test sets reflect a diverse range of regions from Alberta's varied landscape. This approach preserves dataset diversity and simulates real-world conditions where imbalances are common.

This method involves forming small clusters $k_{1i}$ of nearby well patches based on their centroids as illustrated in Figure 3 (a). These small clusters are then grouped into larger, non-intersecting super-clusters $k_{2i}$, with each super-cluster representing a city or larger geographical area. The formation of super-clusters involves calculating a centroid for each $k_{1i}$ cluster based on the centroids of the well patches it contains as illustrated in Figure 3 (b). By clustering wells in this manner, we ensure that $k_{1i}$ clusters group wells from nearby localities together, while $k_{2i}$ clusters group wells from the

Table 3: Information on the numbers of wells represented in the dataset across different states (suspended, abandoned, and active). It also includes domain-specific metadata, such as the mode of operation and the types of fossil fuels extracted, which were used for filtering and quality control.

| Well State | Count | License Status | Mode Short Description | Fluid Short Description |
|---|---|---|---|---|
| Suspended | 55007 | Suspension Issued Amended | All Suspended | Gas, Crude oil, Crude bitumen, Liquid petroleum gas, |
| Abandoned | 54947 | Abandoned Issued Amended | All Abandoned, Abandoned Zone, Junked and Abandoned. | Coalbed methane-coals and other Lith, Coalbed methane-coals only, Shale gas only, Acid gas, |
| Active | 107139 | Issued Amended Re-Entered | Flowing, Pumping,Gas Lift. Abandoned and Re-Entered | CBM and shale and other sources, Shale gas and other sources. |

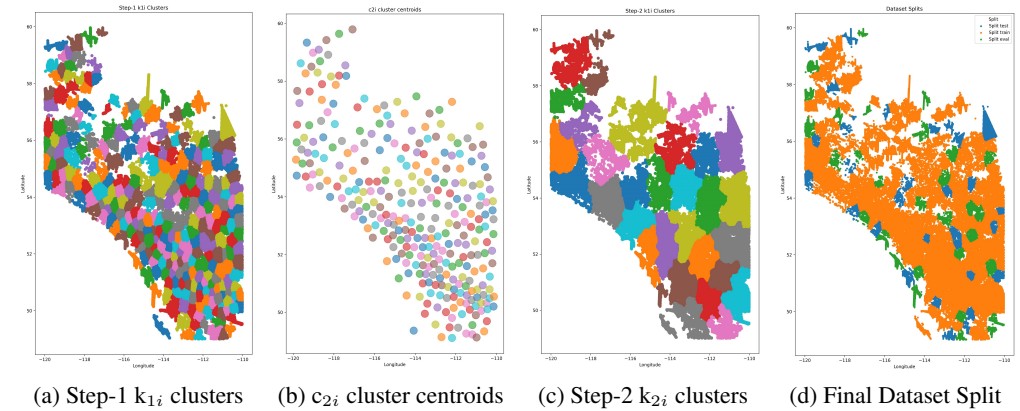

(a) Step-1 $k_{1i}$ clusters   (b) $c_{2i}$ cluster centroids   (c) Step-2 $k_{2i}$ clusters   (d) Final Dataset Split

Figure 3: Illustration of the outcome of applying our dataset splitting algorithm: In Figures (a) to (c), different colors represent various cluster IDs. In Figure (d), blue refers to the training set, orange to the validation set, and green to the test set.

---

**Algorithm 1** Clustering Algorithm for Dataset Splitting

---

$W$: Set of image patches ids containing wells ; $NW$: Set of image patches ids not containing wells

**Input:** $x_i$ represents the $i$-th patch with centroid coordinates $c_i$, where $i \in W$ or $i \in NW$ ;

**Output:** $T_s$: Test Set ; $T_r$: Train Set ; $E_v$: Eval Set ;

**Step 1: Clustering into $M$ Clusters**

Perform K-Means Clustering $k_1(*)$ with $M$ clusters using all centroid coordinates $c_i$, where $i \in W$.

Assign each $i$-th patch into the $m$-th cluster where $m \in \{1,...,M\}$ and $i \in W$: cluster $k_{1i} = k_1(c_i) = m$ and update patches $(x_i, c_i, k_{1i})$

**for** $z \in \{1, \ldots, M\}$ **do**

    $W_{cz} = \{j \in W \mid k_{1j} = z\}$

    Calculate cluster centroids $c_{2j}$ based on values of $c_i$ and update patch: $(x_i, c_i, k_{1i}, c_{2j})$, where $i \in W_{cz}$.

**end for**

**Step 2: Clustering into $N$ Super Clusters**

Let $W_{cc}$ be the set of unique $c_{2j}$ for $j \in W$

Perform K-Means clustering $k_2(*)$ with $N$ clusters using all $c_{2i} \in W_{cc}$.

Assign each $c_{2i} \in W_{cc}$ to $n$-th cluster, where $n \in \{1,..,N\}$ & $k_{2i} = k_2(c_{2i}) = n$.

Update patches $(x_j, c_j, k_{1j}, c_{2j}, k_{2j})$ where $c_{2j} = c_{2i}$ and $j \in W$.

**Step 3: Assigning Patches to Sets**

**for** $z \in \{1, \ldots, N\}$ **do**

    Find all $j$ with $k_{2j} = z$, where $j \in W$ as $W_{fz}$.

    Find unique $k_{1j}$ and count $o_j$ associated with it for $j$ in $W_{fz}$. The, assign $k_{1j}$ with minimum counts as $\min_1$ and $\min_2$.

    For each $i$ in $W_{fz}$, append $i$ to $E_v$ if $k_{1i} = \min_1$, to $T_s$ if $k_{1i} = \min_2$, otherwise to $T_r$.

**end for**

**Step 4: Assigning Non-Well Patches**

**for** each set_counter in $\{E_v, T_s, T_r\}$ **do**

    **for** each unique $k_{1i}$ as $z_i \in set\_counter$ **do**

        Find convex hull radius $r(z_i)$ of area occupied by $c_j$ , where $j \in set\_counter$ & $k_{1j} = z_i$.

        Locate non-well patches $f \in NW$ within radius $r(z_i)$ not in any other cluster; Assign $f$ to cluster $z_i$: $(x_f, c_f, k_{1f}) : k_{1f} = z_i$ .

    **end for**

**end for**

**Step 5: Imbalance Correction**

$T_w$ refers to Count of Well Instances & $T_{nw}$ refers to Count of Non-Well Instances in a Dataset Split

**if** $T_{nw}$ ¿ $T_w$ **then**

    Identify clusters $k_{1j}$ in data split contributing to the imbalance of excess non-well patches, assign to $W_{ic}$

    **for** each $i$ in $W_{ic}$ **do**

        $R(i) = (T_{nw} - T_w) \cdot \frac{\text{Count\_Non\_Wells}(k_{1i})}{\sum \text{Count\_Non\_Wells}(k_{1l}) \text{ where } l \in W_{ic}}$ ; where $R(i)$ is the no. of Samples to be Removed from $i$-th Cluster.

    **end for**

**else**

    Sample non-well patches $x_j : j \in NW$ & $j \notin k_{1j}$.

**end if**

---

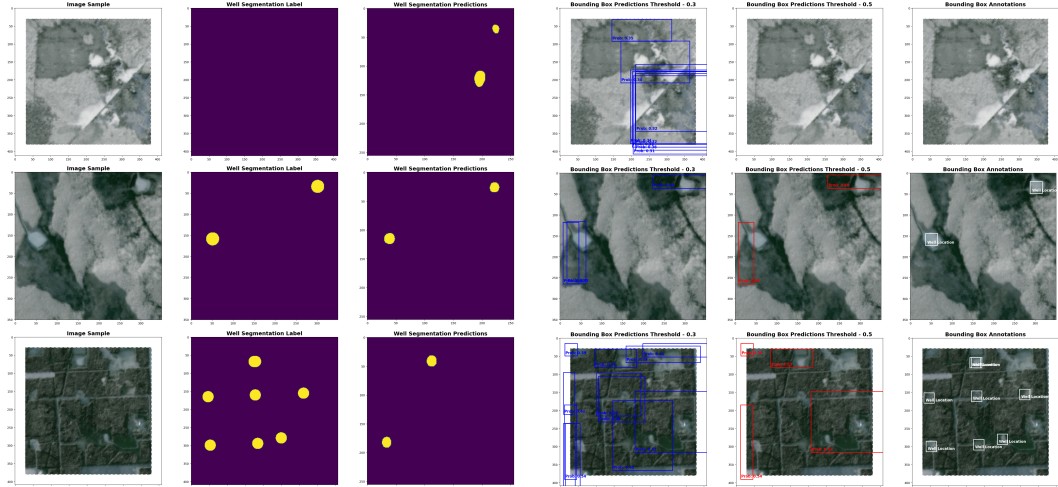

Figure 4: A sample image patch from our dataset includes examples with no wells, two wells, and multiple wells. Additionally, we present qualitative results with predictions generated by our Segmentation U-Net (EfficientNet-B6) and Object Detection FCOS models.

same geographic region as illustrated in Figure 3 (c). Thus, each $k_{2i}$ cluster represents a geographic distribution, with each $k_{1i}$ cluster within it representing a sample of that distribution.

To ensure a diverse and well-distributed evaluation and testing of our machine learning model, we select the $k_{1i}$ clusters with the two fewest well instances from each $k_{2i}$ super-cluster for inclusion in the evaluation and test sets. This approach ensures a diverse representation of the dataset as observed in Figure 3 (d). Moreover, we maintain an equal distribution of well and non-well patches. In cases of imbalance in non-well images, we exclude such patches from the contributing $k_{1i}$ clusters as specified in Algorithm 1. For imbalances in well images, we sample non-well patches that are not part of any other clusters.

The parameters used in constructing the dataset are $M = 300$ and $N = 30$. These were picked heuristically so as to create a well-distributed dataset. Alberta's varied landscape offers a rich environment for creating a comprehensive oil well dataset. Training machine learning models on this extensive dataset improves their robustness and ability to generalize to similar, less-studied regions, thereby supporting well detection and efforts to mitigate global warming. By forming non-overlapping clusters ($k_{1i}$), each with its own well and non-well patches, we minimize the risk of data leakage while ensuring diversity. We also balanced non-well images across clusters to better simulate real-world conditions. This approach helps maintain the diversity of the dataset.

### 3.3 SATELLITE IMAGERY ACQUISITION & LABEL CREATION

We used PlanetScope-4-Band imagery (PBC, 2024) featuring RGB and Near Infrared bands to represent satellite images of the region with a medium resolution of about 3 meters per pixel. PlanetScope, a product of Planet Labs, consists of approximately 130 satellites that can image the entire Earth's land surface daily, collecting up to 200 million sq. km of data each day. We obtained Surface Reflectance imagery, which is offset-corrected, flat-field-corrected, ortho-rectified, visually processed, and radio-metrically corrected. These processes ensure consistency across varying atmospheric conditions and minimize uncertainty in spectral response over time and location, making the data ideal for temporal analysis and monitoring applications.

We choose Planet Labs data over other alternatives since it is updated daily, making it possible to pick a consistent time for all the images, which is important for training dataset consistency. It also provides multispectral imagery (4-band: RGB+Near Infrared), and the Near Infrared band is a useful addition since certain features, like ground depressions indicating well sites, may be more detectable in this band. Lastly, while other alternatives may be limited in remote regions, Planet's

global satellite constellation ensures more consistent coverage. All the imagery we use is made publicly available in the dataset.

To ensure the highest quality, we selected images with no cloud cover. The images were acquired by Planet satellites within a timeframe that aligns with the well-location data from AER. We obtained satellite images for each sample based on geographical coordinates, ensuring an intersection between the actual area of interest and the acquired imagery.

We frame the task of identifying wells as both an object detection and segmentation task since related remote sensing tasks have found both framings to be constructive. For each image patch, as shown in Figure 4, we generated corresponding segmentation maps and object detection annotations for all known wells in the image based on the point labels provided in the AER data. For binary segmentation, we annotated each well site with a circle to match the teardrop shape typical for well sites. We standardized the diameter of a well site to a value of 90 meters (such sites typically range from 70 to 120 meters in diameter). We used the same scale to define bounding boxes in the object detection task, following the COCO (Lin et al., 2014) format for annotations. The overlap in ground truth bounding boxes for some of the wells in Figure 4 and Figure 9 reflects the clustering of multiple wells in densely developed oil and gas sites, where the spatial overlap of wells and infrastructure is common. (Note that this is a characteristic of the data, not a limitation of our quality control strategy.) Additionally, we created multi-class segmentation maps, where each class represents a different state of the well (active, suspended, or abandoned), and included this information in the object detection annotations. (We do not perform multi-class segmentation experiments here, but it is possible that future researchers may find this task useful.)

## 4 BENCHMARK EXPERIMENTS

We train benchmark deep learning models for binary segmentation and object detection tasks. Our focus includes all oil and gas wells, regardless of their operational status, since they exhibit similar footprints and consistent features, making them detectable in satellite imagery.

For both tasks, all models were trained using RGB and Near-Infrared (NIR) channels of the multispectral satellite imagery. We augment images by randomly resizing images to 256×256, ensuring all bounding boxes remain intact for object detection. We then apply horizontal and vertical flipping with a probability of 0.25 each, followed by normalization using channel-wise mean and standard deviation calculated from the training split of the dataset. The hyperparameters we use in these various models represent standard performant settings and are not intended to represent the outcome of hyperparameter optimization.

### 4.1 BINARY SEGMENTATION

We selected well-known baseline models for binary segmentation, encompassing the deep CNN-based approaches U-Net (Ronneberger et al., 2015), PAN(Li et al., 2018), and DeepLabV3+ (Chen et al., 2018) as well as the Transformer-based architectures Segformer (Xie et al., 2021) and UperNet (Xiao et al., 2018).

U-Net (Ronneberger et al., 2015) was chosen for its widespread use as a baseline, offering an effective encoder-decoder architecture for multi-scale feature extraction. PAN (Li et al., 2018) improves multi-scale context with pyramid pooling and attention mechanisms. DeepLabV3+(Chen et al., 2018) was selected for its popularity in remote sensing tasks with its Atrous Convolution and ASPP module for capturing contextual information at various scales. SegFormer (Xie et al., 2021) is a transformer-based architecture designed for semantic segmentation, utilizing self-attention mechanisms for capturing long-range dependencies. UperNet (Xiao et al., 2018) combines UNet (Ronneberger et al., 2015) and PSPNet (Zhao et al., 2016) architectures, featuring a UNet-like structure for multi-scale feature fusion and PSPNet's pyramid pooling module integrated with a Swin Transformer (Liu et al., 2021) backbone for efficient multi-scale processing.

We train all CNN-based models using a ResNet50 (He et al., 2015) backbone, a batch size of 128, and the BCELogits loss function. To fine-tune the model, a cosine annealing scheduler (Loshchilov & Hutter, 2016) is used, which adjusts the learning rate smoothly in a cyclical manner by gradually decreasing it. To evaluate the impact of backbones with larger receptive fields and attention mech-

anisms, we also experimented with additional backbones with U-Net. This included ResNeXt50 (Xie et al., 2016), which enhances feature learning through grouped convolutions; SE-ResNet50 (Hu et al., 2017), which introduces channel-wise attention with Squeeze-and-Excitation blocks; and EfficientNetB6 (Tan & Le, 2019), known for its balanced scaling. For transformer-based models, while both Segformer and UperNet use a Dice loss function and a polynomial learning rate scheduler, Segformer utilizes a mit-b0-ade (Xie et al., 2021) backbone with a batch size of 128, and UperNet employs a Swin Small Transformer with a batch size of 64. All models are optimized using AdamW (Loshchilov & Hutter, 2017) for 50 epochs.

We evaluate the binary segmentation task with respect to IoU, Precision, Recall, and F1-Score. High Precision corresponds to reducing false positives, while high Recall corresponds to reducing false negatives. IoU measures the overlap between predicted and ground truth masks, offering further insight into segmentation accuracy. F1-Score, the harmonic mean of precision and recall, provides a balanced measure considering both false positives and false negatives.

## 4.2 OBJECT DETECTION

For binary object detection, we consider both single-stage, i.e., RetinaNet (Lin et al., 2017), FCOS and SSD, and two-stage CNN-based architectures, i.e. Faster R-CNN (Ren et al., 2015).

RetinaNet (Lin et al., 2017) is a one-stage architecture trained using focal loss, which helps to address class imbalance. It uses a Feature Pyramid Network (FPN) for multi-scale feature extraction and efficient object detection across different scales. Faster R-CNN (Ren et al., 2015) is a two-stage model recognized for its high accuracy. It employs a Region Proposal Network (RPN) for generating region proposals and a separate network for predicting class labels and refining bounding box coordinates. FCOS (Fully Convolutional One-Stage Object Detection) (Tian et al., 2019) directly predicts object locations and categories from feature maps, which is effective for small object detection. SSD (Single Shot MultiBox Detector) (Liu et al., 2015) uses multiple feature maps at different scales, enhancing its accuracy for small objects.

All object detection models are trained using a ResNet50 backbone, except for SSD Lite, which is trained with a MobileNet backbone. The batch size is set to 256 for Faster R-CNN and FCOS and 512 for RetinaNet and SSD Lite. We used a cosine annealing scheduler (Loshchilov & Hutter, 2016) and trained all models for 120 epochs. All models are optimized using the AdamW optimizer (Loshchilov & Hutter, 2017).

For binary object detection model evaluation, we calculate Intersection over Union (IoU) at various thresholds (e.g., $IoU_{0.1}$, $IoU_{0.3}$, $IoU_{0.5}$), which measures how well predicted bounding boxes align with ground truth. IoU is computed by dividing the area of overlap by the area of their union, with higher values indicating better alignment. IoU thresholds define the minimum overlap required for a predicted box to match a ground truth box. (For example, an $IoU_{0.5}$ threshold means a predicted box must have at least 50% overlap with a ground truth box to be considered a correct detection.)

We also assess Mean Average Precision (mAP), including $mAP_{50}$ and $mAP_{50:95}$, measuring the model's precision-recall trade-off and detection accuracy at various IoU thresholds. $mAP_{50}$ measures precision at an IoU threshold of 0.5, while $mAP_{50:95}$ averages precision across IoU thresholds from 0.5 to 0.95. Higher mAP scores reflect better detection accuracy and precision.While higher IoU values indicate better accuracy for individual predictions, mAP provides a broader measure of detection performance by capturing precision across different IoU criteria.

## 4.3 RESULTS & ANALYSIS

Our tasks involve identifying a roughly circular well region with a 90m diameter in real life, which translates to less than 30 pixels in satellite imagery due to resizing and other augmentations. This poses a challenge for machine learning models given the heterogeneous nature of the background, including various similarly shaped and sized features of the natural and built environment. Additionally, vegetation can occlude wells in RGB channels, highlighting the importance of near-infrared imagery for guiding the model. The wells themselves also vary somewhat in shape and can be in various states of disrepair as a result of differing ages and maintenance.

Table 4: Results for the binary segmentation task for a variety of models evaluated over the test set.We report the Intersection over Union (IoU), precision, recall, and F1-score.

| Architecture | Backbone | Params | GFLOPs | IoU | F1 Score | Precision | Recall |
|---|---|---|---|---|---|---|---|
| | ResNet50 | 32.52M | 21.42 | 58±0.5 | 61.9±0.8 | **90.2±2.2** | 62.3±1.6 |
| U-Net | ResNext50 | 32M | 21.81 | 58.2±0.2 | 62.1±0.3 | 88.2±3.5 | 63.6±1.7 |
| | SE_ResNet50 | 35.06M | 20.83 | 58.9±0.7 | 62.9±0.7 | 88.8±1.6 | 64.4±1.4 |
| | EfficientNetB6 | 43.83M | - | **60.4±0.3** | **64.8±0.4** | 87.8±0.4 | 66.3±0.3 |
| PAN | ResNet50 | 24.26M | 17.47 | 57.8±0.8 | 61.5±0.9 | 89.3±1.2 | 61.5±0.9 |
| DeepLabV3+ | | 26.68M | 18.44 | 56.8±0.7 | 60.6±0.7 | 89.4±1.3 | 61.8±1.1 |
| Segformer | mit-b0-ade | 3.72M | 3.52 | 57.6±0.5 | 61.3±0.6 | 82.6±2.9 | 69.2±2.1 |
| UperNet | swin small | 81.15M | 134.20 | 59.9±0.7 | 64.2±0.7 | 80.6±0.5 | **73.1±0.1** |

### 4.3.1 BINARY SEGMENTATION

For the binary segmentation task framing, we train Models (from scratch) using both CNN-based and Transformer-based backbones, considering the prevalent imbalance in the image data due to the small size of wells. Although we did use 3-dimensional, ImageNet initialized weights of the backbone but modified the initial layers afterwards to support 4-dimensional multispectral images.

Among our models, as shown in Table 4, the traditional U-Net with EfficientNetB6 backbone performs the best, with CNN-based models showing the highest IOU of $60.4 \pm 0.3$ and F1-Score of $64.8 \pm 0.4$. While a ResNet50-based backbone achieves the highest Precision of $90.2 \pm 2.2$, indicating more accurate predictions of well instances compared to other models. Precision, which reflects the accuracy of our positive detections compared to the ground truth, is crucial. However, a high recall value ensures the model captures most actual well instances, reducing the risk of missing important information. Thus, the Uper-Net model with the highest recall value of $73.1 \pm 0.1$, which excels at capturing global context information, appears a good candidate for this task given a decent precision score. However, taking into account both precision and recall, U-Net with EfficientNetB6 backbone perform well, suggesting the utility of a larger backbone with a bigger receptive field.

### 4.3.2 BINARY OBJECT DETECTION

Our evaluation, as shown in Table 5, indicates that while all models perform reasonably well in terms of aligning predicted and actual well locations, performance in the object detection task is overall lower than for segmentation – indicating that potentially segmentation is simply a better framing for this task in real-world settings.

The observed gap in performance is likely due to the small size of the wells. It is well-known that single-stage CNN architectures (such as FCOS and SSD) often demonstrate better performance on small object detection than two-stage methods (such as Faster R-CNN), and this aligns with our observations. The exception here is the single-stage method RetinaNet, which, although it has comparable IoU scores, struggles to detect wells accurately. SSD Lite stands out with the highest $IoU_{0.5}$ score of $65.07 \pm 0.03$ and $IoU_{0.3}$ score of $50.3 \pm 0.08$. Whereas all models are quite similar in terms of $IoU_{0.1}$, the highest score by FasterRCNN is $36.79 \pm 1.07$. Thus, SSD Lite and FCOS excel in localization, especially at higher IoU thresholds, while Faster R-CNN is adept at detecting objects with minimal overlap. All models demonstrate low performance in terms of $mAP_{50}$, which assesses precision-recall trade-off and detection accuracy at an IoU threshold of 0.5. FCOS achieves the of $9.67 \pm 1.47$ while SSD Lite achieves a score of $9.76 \pm 0.39$. This may be due to these models not producing region proposals confidently enough, especially in instances with a large number of wells. Whereas over a broader evaluation with $mAP_{50:95}$ which averages precision across IoU thresholds from 0.5 to 0.95. All models apart from RetinaNet provide much better results, with FCOS achieving

Table 5: Results for the object detection task for a variety of models evaluated over the test set. We report the intersection over union (IoU) over thresholds $0.1, 0.3, 0.5$ and the mean average precision (mAP) for both IoU= 0.5 and IoU∈ $[0.5, 0.95]$ thresholds.

| Architecture | Backbone | Params | GFLOPs | $IoU_{0.1}$ | $IoU_{0.3}$ | $IoU_{0.5}$ | $mAP_{50}$ | $mAP_{50:95}$ |
|---|---|---|---|---|---|---|---|---|
| RetinaNet | | 18.87M | 0.93 | 24.58±0.11 | 43.07±0.8 | 59.79±0.36 | 0.18±0.28 | 0.72±1.12 |
| FasterRCNN | ResNet50 | 41.09M | 24.7 | **36.79±1.07** | 46.95±0.66 | 61.29±0.35 | 5.20±1.00 | 19.12±3.41 |
| FCOS | | 31.85M | 25.81 | 34.79±0.99 | 48.51±0.59 | 62.66±0.43 | **9.67±1.47** | **30.46±3.11** |
| SSD Lite | MobileNet | 3.71M | 0.64 | 33.91±0.18 | **50.30±0.08** | **65.07±0.03** | **9.76±0.39** | 25.14±0.66 |

the highest score of $30.46 \pm 3.11$, indicating a decent performance in the identification of well instances.

## 5 LIMITATIONS

We do not envision any significant negative uses of our work. Localization of wells is primarily of interest to the climate change mitigation community and is not, for example, a primary means whereby fossil fuel companies select new locations for drilling. Therefore, we do not believe this dataset is susceptible to dual use risks.

One potential limitation of our work is that we rely on well locations listed by the Alberta Energy Regulator. It is likely that some true well locations are missing in this data, leading to the potential for false negatives in the ground-truth data for this problem. However, it is to be expected that this will not significantly affect the training of algorithms since these labels represent a small fraction of the negative locations in the dataset, and deep learning algorithms are known to be robust to moderate amounts of label noise (see e.g. (Rolnick et al., 2017)). Instead the effect may simply be that the reported test accuracy is actually lower than the true value (due to certain correctly predicted well locations being evaluated as false). We hope to investigate such effects further in future work.

Our dataset is focused on Alberta, because (1) it is a very large region with a significant amount of high-quality labeled data available, (2) it is one of the world's most important locations for oil and gas production, so identifying wells in Alberta is of immediate impact. Future works may wish to assess the capacity for few- or zero-shot transfer learning from Alberta to other regions with a high expected concentration of abandoned wells, including the Appalachian and Mountain West regions of the United States, as well as a number of former Soviet states.

## 6 CONCLUSION

In this paper, we present the first large-scale benchmark dataset aimed at identifying oil and gas wells, with a focus on abandoned and suspended wells, which are a significant source of greenhouse gases and other pollutants. We combine high-resolution imagery, an extensive database of well locations, and expert verification to create the Alberta Wells Dataset. We frame well identification both in terms of object detection and binary segmentation and evaluate the performance of a wide range of popular deep learning methods on these tasks. We find that the UNet model (with a Efficient-NetB6 backbone), in particular, represents the most promising baseline for the binary segmentation task, while for object detection, all models demonstrate more mixed results, with Single Stage Models (such as FCOS and SSD) providing a relatively promising baseline. These results show that the Alberta Wells Dataset represents both a challenging as well as a societally impactful set of tasks.

The value added by the dataset is twofold. First, most global fossil fuel-producing regions do not have databases of well locations comparable to that provided by AER. Alberta's varied landscape provides an ideal setting for developing algorithms for the detection of wells, which can then be used directly in other locations or fine-tuned. Second, even in Alberta, the list developed by AER is not comprehensive, and many abandoned wells are believed to be missing. Our work can provide candidate locations for domain experts to examine so as to determine the true number and locations of abandoned wells in Alberta.

We hope that our work may be of use to policymakers and other stakeholders involved in climate action and environmental protection according to the following envisioned steps:

- Use the Alberta Wells Dataset to train algorithms for pinpointing well locations.
- Run these algorithms at scale across a broader region of interest, comparing against any existing databases to identify those wells that may be undocumented.
- Flag abandoned wells for plugging, prioritizing those identified as super-emitters.

We believe that the scalability of machine learning tools for remote sensing will make them an invaluable tool in pinpointing and mitigating the global environmental impact of abandoned oil and gas wells.

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

# A ADDITIONAL EXPERIMENTS

## A.1 IMPACT OF NEAR INFARED MULTI-SPECTRAL IMAGERY BAND (RGB V/S RGB+NIR)

The inclusion of the Near-Infrared (NIR) band significantly improves both object detection and segmentation performance over the standard RGB modality as illustrated in Tables 6 and 7.

In object detection (Table 7), RGB+NIR achieves higher Intersection over Union (IoU) scores across all thresholds (0.1, 0.3, and 0.5) and a considerable increase in mAP@50 ($9.67 \pm 1.47$ vs. $5.7 \pm 3.65$) and mAP@50:95 ($30.46 \pm 3.11$ vs. $20 \pm 10.4$). Similarly, in segmentation (Table 6), RGB+NIR shows superior performance in IoU ($58 \pm 0.5$ vs. $56.6 \pm 0.44$), F1 Score ($61.9 \pm 0.8$ vs. $60.5 \pm 0.35$), and Precision ($90.2 \pm 2.2$ vs. $87 \pm 1.4$), while maintaining a slightly lower Recall ($62.3 \pm 1.6$ vs. $62.54 \pm 0.13$).

These improvements can be attributed to the enhanced spectral information provided by the NIR band, which is particularly effective in detecting features such as ground depressions that may indicate well sites. These features are often more distinguishable in the NIR spectrum, leading to better performance in both tasks.

Table 6: Results for the binary segmentation task for U-Net Model with ResNet50 backbone evaluated over the test set for multiple input modality. We report the Intersection over Union (IoU), precision, recall, and F1-score.

| Modality | GFLOPs | Params | IoU | F1 Score | Precision | Recall |
|---|---|---|---|---|---|---|
| RGB+NIR | 21.42 | 32.52M | **58.00±0.50** | **61.9±0.80** | **90.20±2.20** | 62.30±1.60 |
| RGB | 21.32 | 32.52M | 56.60±0.44 | 60.50±0.35 | 87.00±1.40 | **62.54±0.13** |

Table 7: Results for object detection task for the FCOS Model with ResNet50 backbone evaluated over the test set for multiple input modality. We report the intersection over union (IoU) over thresholds $0.1, 0.3, 0.5$ and the mean average precision (mAP) for both IoU= $0.5$ and IoU∈ $[0.5, 0.95]$ thresholds.

| Modality | GFLOPs | Params | $IoU_{0.1}$ | $IoU_{0.3}$ | $IoU_{0.5}$ | $mAP_{50}$ | $mAP_{50:95}$ |
|---|---|---|---|---|---|---|---|
| RGB+NIR | 25.81 | 31.85M | **34.79±0.99** | **48.51±0.59** | **62.66±0.43** | **9.67±1.47** | **30.46±3.11** |
| RGB | 25.71 | 31.85M | 32.39±2.88 | 46.80±2.07 | 61.23±1.58 | 5.7±3.65 | 20.00±10.40 |

## A.2 ADDITIONAL EXPERIMENTS: CONVNEXT BACKBONE

UperNet is a robust semantic segmentation framework that integrates multi-scale features using a Feature Pyramid Network and a refined decoder to capture both global and local context, making it effective for complex segmentation tasks. ConvNeXt, a modern convolutional backbone, enhances feature extraction through advanced architectural refinements inspired by transformer models.

As shown in Table 8, ConvNeXt-Base achieves an IoU of 59.89, F1 score of 64.04, precision of 81.71, and recall of 72.05, making it a competitive choice for remote sensing image analysis. Compared to the architectures presented in Table 4, such as U-Net with EfficientNetB6 (IoU: 60.4, F1 score: 64.8) and UperNet with Swin Small (IoU: 59.9, F1 score: 64.2), ConvNeXt-Base offers a comparable balance of accuracy and recall, particularly excelling in recall at 72.61, which is critical for identifying all relevant features in real-world scenarios. Additionally, ConvNeXt Small provides a trade-off between computational efficiency and model size, requiring fewer GFLOPs and parameters compared to UperNet with Swin Small while achieving similar performance.

Table 8: Results for the binary segmentation task for UperNet Model with ConvNexT backbones evaluated over the test set for multiple input modality. We report the Intersection over Union (IoU), precision, recall, and F1-score.

| Backbone | GFLOPs | Params | IoU | F1 Score | Precision | Recall |
|---|---|---|---|---|---|---|
| ConvNexT-Small | 128.29 | 81.76M | 59.47 | 63.60 | 81.01 | 71.82 |
| ConvNexT-Base | 146.27 | 121.99M | **59.89** | **64.04** | **81.71** | **72.05** |

Table 9: Performance comparison of U-Net Model with ResNet50 Backbone for Binary Segmentation trained on active wells only versus all well types over Test Set. The table highlights metrics (IoU, F1 score, precision, and recall) demonstrating the importance of incorporating all well types in the training dataset for improved generalization and balanced performance.

| Metric | Train Set (Well Type Label Present) | Test Set |
|---|---|---|
| IoU | Active (I) | 0.502 |
| | All (I+II+III) | **0.576** |
| F1 Score | Active (I) | 0.503 |
| | All (I+II+III) | **0.614** |
| Precision | Active (I) | **0.998** |
| | All (I+II+III) | 0.913 |
| Recall | Active (I) | 0.502 |
| | All (I+II+III) | **0.614** |

### A.3 BENEFITS OF USING MULTIPLE WELL TYPES

The results in Table 9 emphasize the necessity of including all well types (active, suspended, and abandoned) in the training dataset to ensure comprehensive detection across diverse scenarios. Training exclusively on active wells significantly underperforms in mixed-type contexts (IoU: 0.502), indicating poor generalization when all well types are present. Conversely, training on all well types improves the model's ability to handle real-world heterogeneity, as reflected by a higher IoU (0.576), F1 score (0.614), and recall (0.614) for the test set. The enhanced recall demonstrates the model's capability to identify a broader range of wells, crucial for environmental monitoring, where missing even a single abandoned well could result in unaddressed methane emissions or groundwater contamination. Precision for active-only training (0.998) is higher than model trained on all well types (0.913), but its inability to detect all wells in a image (due to low recall) limits its applicability. Therefore, incorporating all well types in training ensures balanced, reliable performance, allowing for accurate detection and classification in diverse and realistic contexts.

## B DATASET AND CODE

### B.1 DATASET

The dataset is currently hosted in Dropbox for anonymity reasons and can be accessed here:
AWD Dataset

- Example visualization of dataset samples, including the spectral bands of each image and the corresponding labels: Visualizations
- Compressed training set: Train.tar.gz
- Compressed validation set: Validation.tar.gz
- Compressed test set: Test.tar.gz
- Dataset license: License.txt

### B.2 CROISSANT METADATA

The Croissant metadata can be accessed from here: Croissant metadata record (AWD)

Our dataset is comprised of Hierarchical Data Format (HDF5) files with a multi-level hierarchy. The Croissant metadata format does not currently support describing the structure within each HDF5 file, as noted in a GitHub issue.

Therefore, we provide Croissant dataset metadata that includes only dataset-level information and resources, excluding RecordSets that require data from HDF5 files. We will update the metadata once Croissant supports the HDF5 format.

Additionally, we provide another documentation framework, Datasheets for Datasets Gebru et al. (2021), described in Section G.

We also describe the dataset structure and the structure of data in Hierarchical Data Format (HDF5) files in detail in Sections D.1, D.2 and D.3.

### B.3 CODE REPOSITORY

The code repository with benchmark experiments and visualizations of samples can be accessed here: awd_benchmark

## C HOSTING, LICENSING, AND MAINTENANCE PLAN

### C.1 HOSTING & MAINTENANCE

Once the dataset is made public, we plan to host it on Zenodo.

### C.2 DATA LICENSING

The AWD Dataset is released under a Creative Commons Attribution-NonCommercial 4.0 International (CC BY-NC 4.0) License (`https://creativecommons.org/licenses/by-nc/4.0/`).

The satellite imagery for this project was acquired through Planet Labs' PBC (2024) Education & Research license, which allows the use of the data in publications and the creation of derivative products related to those publications. However, the raw imagery cannot be shared publicly. To adhere to these guidelines, we provide the data in HDF5 format, with the satellite imagery preprocessed to produce a derived product represented as a numpy array from Raster Vector. This process removes all geographic metadata.

This data is for academic use only and should not be used commercially. Proper credit to the current authors, Planet Labs PBC (2024), and the Alberta Energy Regulator AER (2024) is required when using this data.

## D DATASET INFORMATION

The purpose of this dataset is to assist in training deep learning systems to identify oil and gas wells, including abandoned, suspended, and active ones. This will enable the detection of wells in a specific area, allowing comparison with government records. If discrepancies are found, experts can conduct further investigations, which can possibly lead to the discovery of an abandoned or suspended device that might not be present in government records.

### D.1 DATASET STRUCTURE

We provide training, validation, and testing sets, split using our proposed algorithm (as described in Section 3.2 of the main paper) to create a well-distributed dataset.

The proposed method aims to create smaller regions of well concentration by clustering the centroids of patches. These regions are designed to be (a) mutually non-intersecting, (b) part of a larger geographic region by clustering the centroids of the initial clusters, and (c) containing a similar distribution of non-well patches within the same region.

This approach ensures that the training, validation, and test sets include representations from all geographic regions, providing a diverse and comprehensive evaluation. Thus, the dataset represents various geographical regions and offers a diverse benchmark for evaluation and testing.

Each dataset split is saved in an HDF5 format file, structured as described in the following sections, and then compressed into a .tar.gz file for faster transfer. Details on the number of samples in each set and the size of the dataset, both original and compressed, are presented in Table 3.

## D.2 DATASET FILE DIRECTORY STRUCTURE

The following directory structure is used for each dataset file being stored in a Hierarchical Data Format 5 (HDF5) file:

```
<Train/Test/Val>Set.h5
    |---image
      |---<sample_name>
          |---Satellite Image (Multispectral Rasterio Image Data)
          |---Meta Data of <sample_name>
    |---label
      |---binary_seg_maps
          |---<sample_name>
              |---Binary Segmentation Map (Rasterio Image Data)
      |---multi_class_seg_maps
          |---<sample_name>
              |---Multiclass Segmentation Map (Rasterio Image Data)
      |---bounding_box_annotations
          |---<sample_name>
              |---Bounding Box JSON Data (COCO Format)
    |---author:Anonymous Author(s)
    |---description: Alberta Wells Dataset:
                     Pinpointing Oil and Gas Wells
                     from Satellite Imagery
```

## D.3 STRUCTURE OF DATASET DIRECTORY

To enhance the efficiency of the data loader, we split the larger .h5 dataset into smaller .h5 files, each corresponding to a unique sample (image patch). By splitting the dataset in such a manner, we are able to improve the speed per iteration of the dataloader by over 100%.

This results in the following data structure:

```
<Sample_Id>.h5
    |---image
      |---Satellite Image (Multispectral Rasterio Image Data)
      |---Meta Data
    |---label
      |---binary_seg_maps
          |---Binary Segmentation Map (Rasterio Image Data)
      |---multi_class_seg_maps
          |---Multiclass Segmentation Map (Rasterio Image Data)
      |---bounding_box_annotations
          |---Bounding Box JSON Data (COCO Format)
    |---author:Anonymous Author(s)
    |---description: Alberta Wells Dataset:
                     Pinpointing Oil and Gas Wells
                     from Satellite Imagery
```

## D.4 DATASET SIZE & DISTRIBUTION OF SAMPLES

Our dataset comprises over 94,000 patches of satellite imagery containing wells, with a total of 188,000 patches sourced from Planet Labs (PBC, 2024), covering more than 213,000 individual wells. Details about the distribution of the number of patches, wells present, and dataset split sizes are provided in Table 10, with the distribution of the number of wells per sample being described in Table 11. We also include an equal number of images that contain no wells in each dataset split. The distribution of wells per sample, along with the corresponding number of wells and the breakdown of well types, is illustrated in Figure 6 and detailed in Tables 12, 13, and 14 . The geographic distribution of wells in the dataset can be visualized in Figure 5.

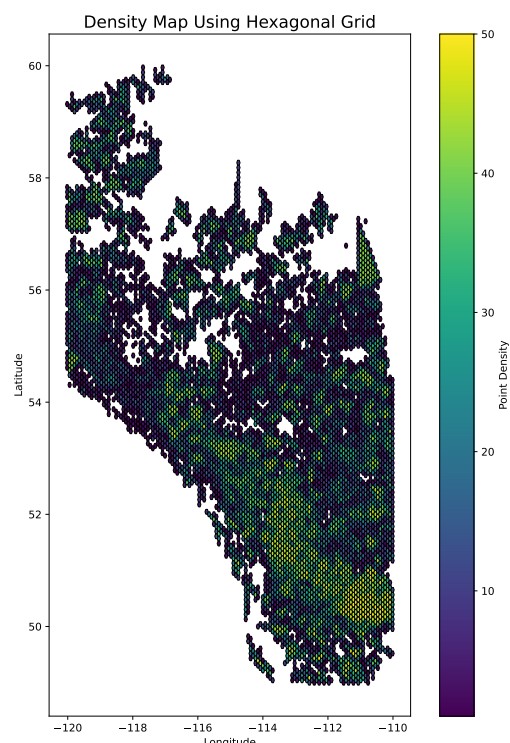

Figure 5: Density map of wells in the Alberta Wells Dataset.

Table 10: Dataset statistics represented across the various splits of the dataset.

| Dataset Split | No of Samples | No of Wells in Split | Original HDF5 File Size (in Gb) | Compressed .tar.gz File Size (in Gb) |
|---|---|---|---|---|
| Train | 167436 | 194231 | 322 | 100 |
| Validation | 9463 | 8243 | 19 | 5.7 |
| Test | 11789 | 10973 | 24 | 7.1 |
| Total | 188688 | 213447 | 365 | 112.8 |

Table 11: The distribution of individual wells in positive samples from the dataset. We also include an equal number of images that contain no wells in each dataset split.

| No of Wells in a Sample | Frequency of Well Instances in a Sample | | |
|---|---|---|---|
| | Training Split | Validation Split | Test Split |
| 1 | 44299 | 3393 | 4128 |
| 2 - 3 | 25378 | 979 | 1242 |
| 4 - 5 | 7899 | 190 | 328 |
| 6 - 10 | 4927 | 123 | 227 |
| 11 - 15 | 751 | 23 | 38 |
| 16 - 25 | 333 | 11 | 19 |
| 26 - 35 | 67 | 10 | 2 |
| 36 - 55 | 45 | 3 | 0 |
| 56 - 75 | 18 | 0 | 0 |
| 76 - 125 | 1 | 0 | 0 |
| Total | 83718 | 4732 | 5984 |

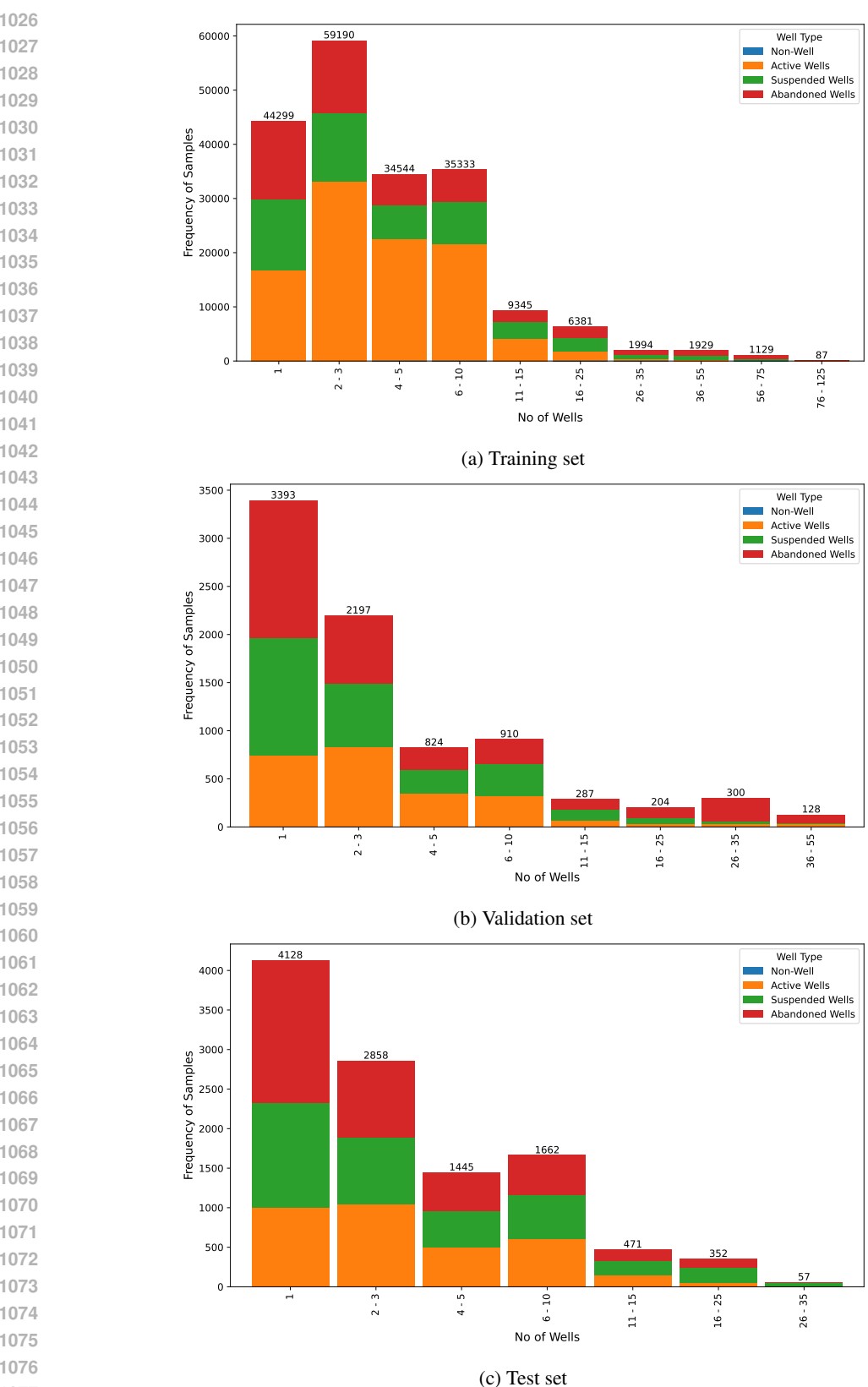

Figure 6: Distribution of the number of individual wells and the proportion of well types (active, suspended, and abandoned) in positive samples from the dataset. We also include an equal number of images with no wells at all.

Table 12: Test set statistics showing the distribution of image samples by the number of wells per image and the breakdown of well types (active, suspended, and abandoned).

| No of Wells in a Image Sample (S) | Count (Image Samples) | Test Set | | | |
|---|---|---|---|---|---|
| | | Distribution of Well Type in Samples | | | |
| | | Total Wells | Active Wells | Suspended Wells | Abandoned Wells |
| 1 | 4128 | 4128 | 999 | 1325 | 1804 |
| 2 - 3 | 1242 | 2858 | 1042 | 844 | 972 |
| 4 - 5 | 328 | 1445 | 495 | 464 | 486 |
| 6 - 10 | 227 | 1662 | 604 | 555 | 503 |
| 11 - 15 | 38 | 471 | 144 | 181 | 146 |
| 16 - 25 | 19 | 352 | 56 | 184 | 112 |
| 26 - 35 | 2 | 57 | 0 | 56 | 1 |
| | | 10973 | 3340 | 3609 | 4024 |

Table 13: Validation Set statistics showing the distribution of image samples by the number of wells per image and the breakdown of well types (active, suspended, and abandoned).

| No of Wells in a Image Sample (S) | Count (Image Samples) | Validation Set | | | |
|---|---|---|---|---|---|
| | | Distribution of Well Type in Samples | | | |
| | | Total Wells | Active Wells | Suspended Wells | Abandoned Wells |
| 1 | 3393 | 3393 | 743 | 1225 | 1425 |
| 2 - 3 | 979 | 2197 | 833 | 654 | 710 |
| 4 - 5 | 190 | 824 | 346 | 248 | 230 |
| 6 - 10 | 123 | 910 | 323 | 331 | 256 |
| 11 - 15 | 23 | 287 | 67 | 114 | 106 |
| 16 - 25 | 11 | 204 | 32 | 63 | 109 |
| 26 - 35 | 10 | 300 | 33 | 22 | 245 |
| 36 - 55 | 3 | 128 | 29 | 14 | 85 |
| | | 8243 | 2406 | 2671 | 3166 |

Table 14: Train Set statistics showing the distribution of image samples by the number of wells per image and the breakdown of well types (active, suspended, and abandoned).

| No of Wells in a Image Sample (S) | Count (Image Samples) | Train Set | | | |
|---|---|---|---|---|---|
| | | Distribution of Well Type in Samples | | | |
| | | Total Wells | Active Wells | Suspended Wells | Abandoned Wells |
| 1 | 44299 | 44299 | 16715 | 13116 | 14468 |
| 2 - 3 | 25378 | 59190 | 33099 | 12706 | 13385 |
| 4 - 5 | 7899 | 34544 | 22456 | 6321 | 5767 |
| 6 - 10 | 4927 | 35333 | 21522 | 7796 | 6015 |
| 11 - 15 | 751 | 9345 | 4076 | 3136 | 2133 |
| 16 - 25 | 333 | 6381 | 1781 | 2544 | 2056 |
| 26 - 35 | 67 | 1994 | 376 | 791 | 827 |
| 36 - 55 | 45 | 1929 | 172 | 777 | 980 |
| 56 - 75 | 18 | 1129 | 86 | 345 | 698 |
| 76 - 125 | 1 | 87 | 11 | 63 | 13 |
| | 83718 | 194231 | 100294 | 47595 | 46342 |

## D.5 PLANETSCOPE SATELLITE IMAGERY

For our experiments, we selected a 4-band (RGBN) satellite imagery product (ortho_analytic_4b_sr) from Planet Labs (PBC, 2024) as illustrated in Figure 7. This product uses Planet's PSB.SD instrument, which features a telescope with a larger 47-megapixel sensor and is designed to be interoperable with Sentinel-2 imagery in several bands. The frequency of each band of image is described in Table 15. The instrument provides a frame size of 32.5 km x 19.6 km, an image capture capacity



(a) A sample patch with Bbox annotations and the corresponding imagery in its different spectral bands.



(b) A sample patch with its segmentation labels (binary and multi-class) and bounding box annotations.

Figure 7: A Sample Patch from the Evaluation Set with 2 active wells.

of 200 million km²/day, and an imagery bit depth of 12-bit, with a ground sample distance (nadir) ranging from 3.7 m to 4.2 m.

The satellite images are corrected for atmospheric conditions and spectral response consistency. These multispectral products are tailored for monitoring in agriculture and forestry, offering precise geolocation and cartographic projection. They are ideal for tasks such as land cover classification, with radiometric corrections ensuring accurate data transformation.

Table 15: The Frequency of Each Spectral Band of a Planetscope PS.SD acquired Image

| Band of Image | Frequency (in nm) of Spectral Band |
|---|---|
| Band 1 = Blue | 465 - 515 |
| Band 2 = Green | 547 - 585 |
| Band 3 = Red | 650 - 680 |
| Band 4 = Near-infrared | 845 - 885 |

## D.6 META DATA DESCRIPTION

Each dataset sample is accompanied by metadata, including the sample name (sample ID in string format), the presence of a well in the sample, the number of wells in the sample, and whether a well of a specific category is present in the sample. Table 16 provides an illustration of metadata associated with a sample.

Table 16: Sample of Meta-Data Associated with each Instance in the Dataset

| Meta-Data Attribute Name | Value |
|---|---|
| Sample_Name | eval_6934 |
| wells_present | True |
| no_of_wells | 10 |
| Abandoned_well_present | True |
| Active_well_present | True |
| Suspended_well_present | True |

## D.7 LABEL DATA DESCRIPTION

For our experiments, we create single-channel segmentation maps, which are binary maps used to locate instances of wells. We also generate multi-class segmentation maps, where each class denotes a well in an active, abandoned, or suspended state. Furthermore, we provide COCO format

object detection labels for wells. In both segmentation and detection labels, we represent various states with class IDs as 'Active': 1, 'Suspended': 2, 'Abandoned': 3. To maintain consistency, we standardize the diameter of a well site to 90 meters (typically ranging from 70 to 120 meters) when annotating, resulting in a 30-pixel diameter in the labels. Figure 7(b) illustrates image patches with their corresponding labels, Figure 7(a) illustrates various spectral bands present in an image alongside the original image with bounding box annotations for reference and an example of a bounding box label in COCO format is shown below.

Sample of Bounding Box Annotation:

```
[
    {
        'id': 0,
        'image_id': 'eval_7028',
        'category_id': 1,
        'bbox': [46, 145, 29, 29],
        'iscrowd': 0
    },
    {
        'id': 1,
        'image_id': 'eval_7028',
        'category_id': 2,
        'bbox': [45, 127, 29, 29],
        'iscrowd': 0
    }
]
```

## E    DATASET SAMPLES ILLUSTRATION

Samples from the dataset, covering various scenarios, are shown in Figures 8 and 9.

## F    CHALLENGES FOR ML COMMUNITY

The Alberta Wells Dataset presents several intriguing challenges for machine learning. Key issues include an imbalanced data distribution, with fewer instances of areas with multiple wells compared to those with single or two wells, and the visual similarity among active, suspended, and abandoned wells, which can confuse standard models. Additionally, varying spatial relationships in the imagery due to varying geography create difficulties for off-the-shelf models. Noise in annotations, even after data quality control and cleaning—such as misclassified wells—further complicates the task. Despite these challenges, the dataset's large scale and geographical diversity, covering over 213,000 wells, offer significant opportunities for developing robust and generalizable ML models for monitoring oil and gas infrastructure.

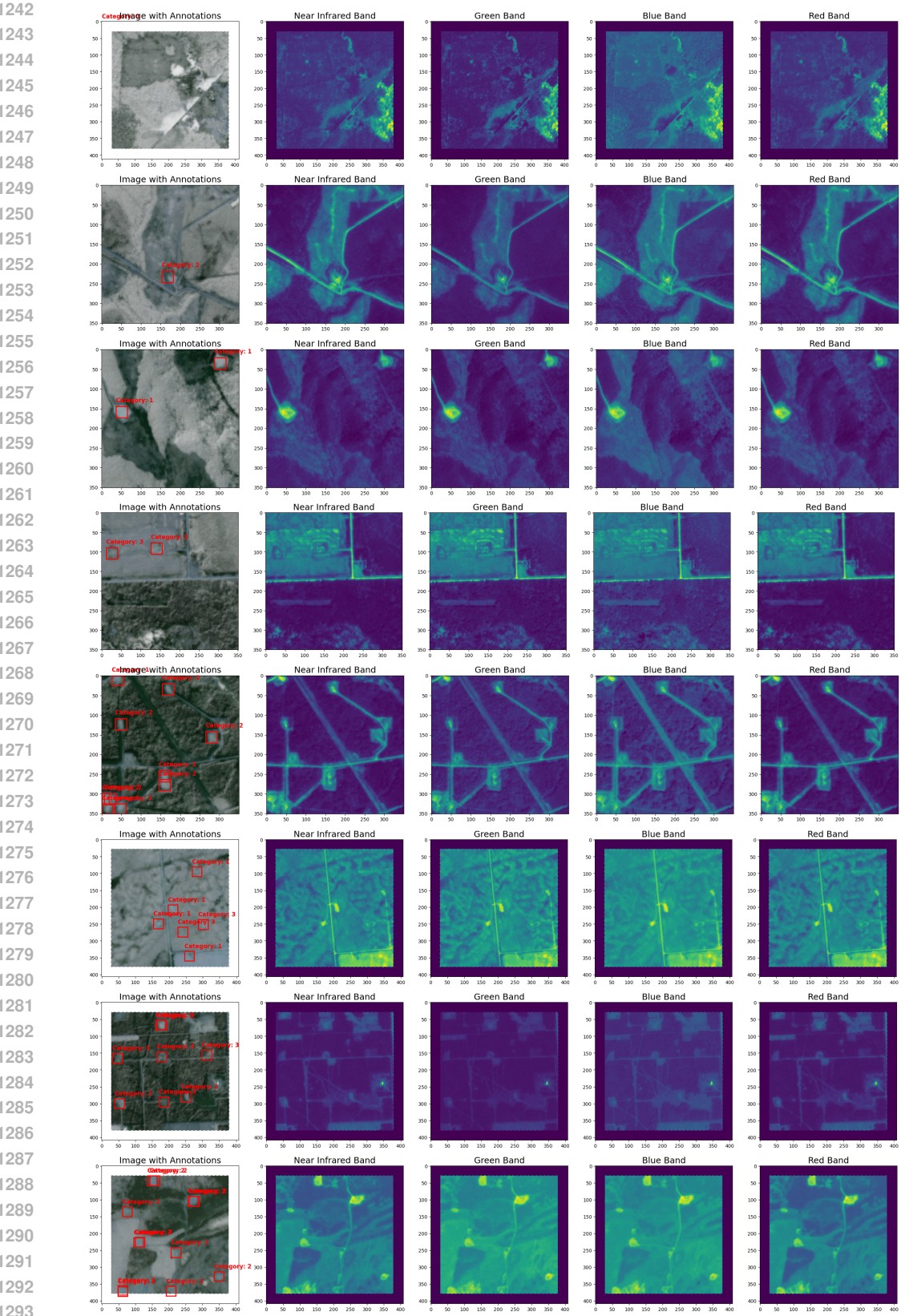

Figure 8: Qualitative results from the dataset illustrate the diverse distribution of wells in dataset samples, including Bbox annotations and corresponding imagery in different spectral bands.

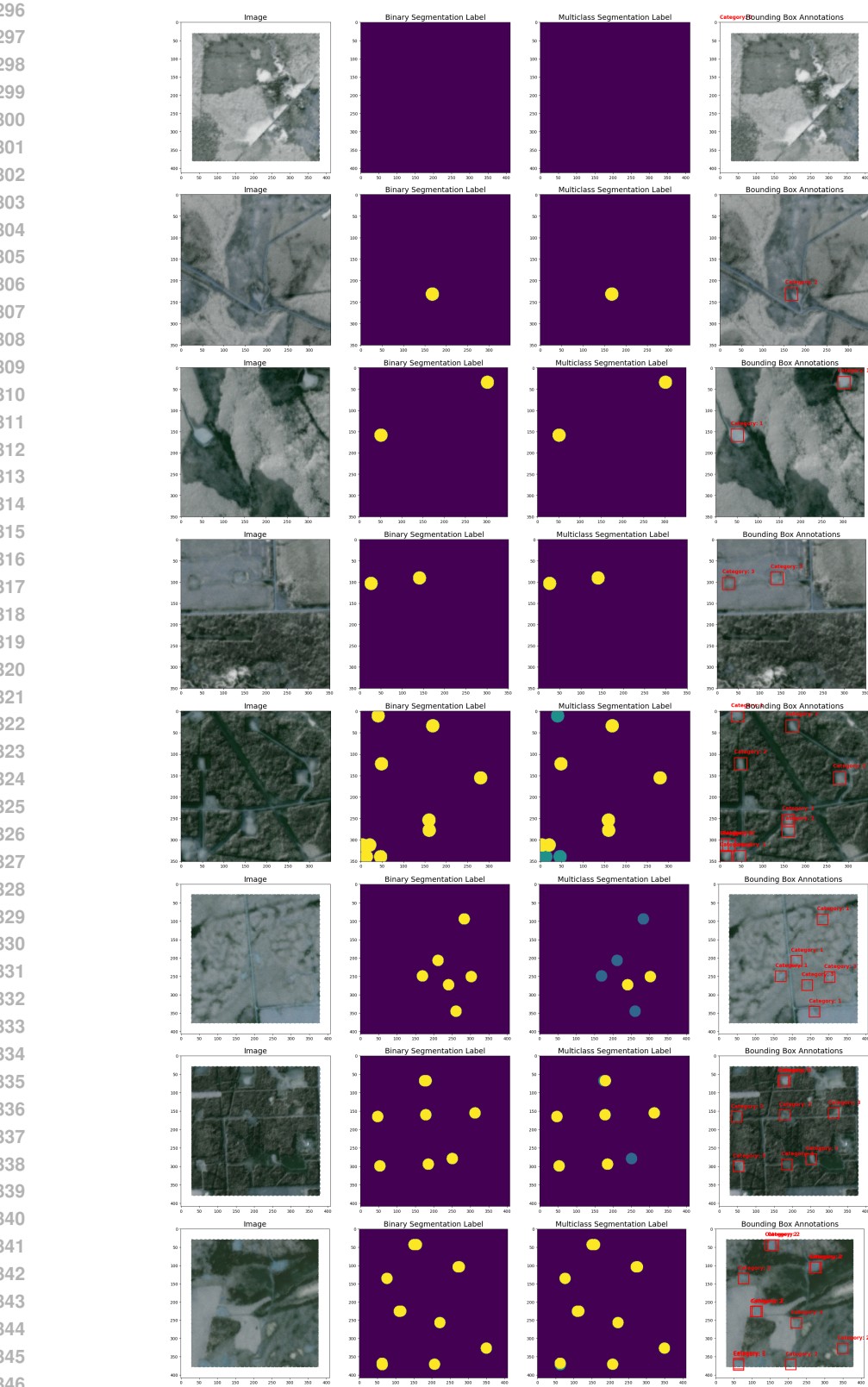

Figure 9: The qualitative results from the dataset showcase the varied distribution of wells in dataset samples, with their corresponding segmentation labels (binary and multi-class) and Bbox annotations.

# G DOCUMENTATION FRAMEWORKS: DATASHEET FOR DATASETS

## G.1 MOTIVATION

1. **For what purpose was the dataset created?** Was there a specific task in mind? Was there a specific gap that needed to be filled? Please provide a description.
   The Alberta Wells Dataset (AWD) was created to identify oil and gas wells—whether abandoned, suspended, or active—using high-resolution (3m/px) multi-spectral satellite imagery. While the issue of detecting oil and gas wells has been addressed by several authors, existing datasets are typically small (500-5,000 samples) and limited to specific regions, often including only active wells. This limitation reduces their effectiveness in identifying abandoned or suspended wells. The AWD aims to fill this gap in the literature by offering a comprehensive dataset with over 188,000 samples (including over 94,000 samples containing wells) from PlanetLabs satellite imagery, encompassing more than 213,000 individual wells.

2. **Who created the dataset (e.g., which team, research group) and on behalf of which entity (e.g., company, institution, organization)?**
   The raw data is sourced from the Alberta Energy Regulator (AER), specifically from the monthly AER ST37 publication. This dataset includes comprehensive details about all reported wells in Alberta, such as their geographic location, mode of operation, license status, and the type of product extracted, among other attributes. The data is provided in shapefile format along with accompanying metadata. However, the dataset cannot be used directly because the license status or mode of operation often does not reflect the well's actual status. Therefore, the authors include domain experts from `Anonymous`, who specialize in field measurements of methane and air pollutant emissions from oil, gas, and urban systems, as well as in the geospatial and statistical data analysis of emissions and energy infrastructure, to ensure the quality of the dataset.

3. **Who funded the creation of the dataset?** If there is an associated grant, please provide the name of the grantor and the grant name and number.
   This project was funded by `Anonymous`.

## G.2 COMPOSITION

- **What do the instances that comprise the dataset represent (e.g., documents, photos, people, countries)?** Are there multiple types of instances (e.g., movies, users, and ratings; people and interactions between them; nodes and edges)? Please provide a description.
  We provide a dataset file stored in Hierarchical Data Format 5 (HDF5, i.e., a .h5 file), which contains multispectral 4-band RGBN satellite images in raster format and data labels with both identified by unique instance names. These satellite images, acquired from Planet Labs, have a resolution of 3 meters per pixel and include corresponding metadata. The metadata contains information about the number and types of wells present in a patch. For data labels, we offer binary segmentation maps, multi-class segmentation maps (each class representing a well in an active, abandoned, or suspended state), and COCO format object detection labels. The images were taken from the province of Alberta, Canada, with each satellite imagery patch representing a square with a side length of 1050 meters (1.05 km), covering an area of 1.025 square kilometers. The entire dataset spans over 193,000 square kilometers.

- **How many instances are there in total (of each type, if appropriate)?**
  The proposed dataset comprises 188,688 instances, of which 94,344 contain one or more wells, totaling 213,447 well points. Each instance includes corresponding multispectral satellite imagery, segmentation maps (both binary and multi-class, with classes indicating active, suspended, or abandoned states), and bounding box annotations with the state of operations as the object class ID in COCO format. We standardized the diameter of a well site to 90 meters (typically ranging from 70 to 120 meters) for creating annotations, resulting in a diameter of 30 pixels in the labels. More details about the distribution of wells in each split are provided in the supplementary materials as well as the main paper.

- **Does the dataset contain all possible instances or is it a sample (not necessarily random) of instances from a larger set?** If the dataset is a sample, then what is the larger

set? Is the sample representative of the larger set (e.g., geographic coverage)? If so, please describe how this representativeness was validated/verified. If it is not representative of the larger set, please describe why not (e.g., to cover a more diverse range of instances because instances were withheld or unavailable).

The AWD Dataset is based on the AER ST37 monthly status data of wells in the Alberta region of Canada. It includes wells that are in active, suspended, or abandoned states of operation. To ensure the dataset's quality, the authors with appropriate domain expertise conducted extensive quality control, filtering, and duplicate removal. This process was necessary because the full dataset included cases of well sites being restored and reclaimed, as well as various duplicates, noise, and data on other types of wells involving different natural resources. Therefore, the AWD Dataset, which includes multi-spectral satellite imagery, segmentation, and detection labels, is constructed from a refined subset of the original AER ST37 data, specifically targeting oil and gas wells that can be precisely identified.

- **What data does each instance consist of?** "Raw" data (e.g., unprocessed text or images) or features? In either case, please provide a description.

  Each Image instance in our dataset, formatted in HDF5, contains satellite imagery represented as a numpy array from Raster Vector. We preprocessed this imagery by reprojecting it to the EPSG 32611 coordinate reference system and removed all geographic metadata, such as image bounds and coordinates, from the shared data. However, we do provide attributes like Sample Name, wells present, no of wells, Abandoned well present, Active well present, and Suspended well present. We utilized Planet Labs' 4-band (RGBN) satellite imagery product (ortho_analytic_4b_sr), which incorporates the latest PSB.SD instrument with a 47-megapixel sensor. Each satellite imagery patch acquired represents a square with a side length of 1050 meters (1.05 km), covering an area of 1.025 square kilometers. The entire dataset spans over 193,000 square kilometers.

- **Is there a label or target associated with each instance?** If so, please provide a description.

  There are three types of labeled data for each image: binary segmentation maps (in Rasterio Image format) indicating the presence or absence of oil and gas wells, multiclass segmentation maps (also in Rasterio Image format) potentially identifying various classes of objects, and bounding box annotations (in COCO format) specifying the location and size of objects, such as wells, within the image. These components together form a comprehensive dataset suitable for training and evaluating machine learning models for tasks like object detection and segmentation in satellite imagery, particularly focused on pinpointing oil and gas wells in Alberta

- **Is any information missing from individual instances?** If so, please provide a description, explaining why this information is missing (e.g., because it was unavailable). This does not include intentionally removed information but might include, e.g., redacted text.

  The satellite imagery used in this project was obtained under Planet Labs' (PBC, 2024) Education & Research license, which prohibits sharing raw satellite imagery. We re-projected the raw data to EPSG:32611 using the nearest resampling method and removed all geographic metadata, such as image bounds and coordinates, from the shared data imagery to create a derived product that complies with the license terms.

- **Are relationships between individual instances made explicit (e.g., users' movie ratings, social network links)?** If so, please describe how these relationships are made explicit.

  N/A

- **Are there recommended data splits (e.g., training, development/validation, testing)?** If so, please provide a description of these splits, explaining the rationale behind them.

  The dataset we propose comprises more than 94,000 patches of satellite imagery containing wells, totaling 188,000 patches sourced from Planet Labs. This dataset covers over 213,000 individual wells. To ensure a balanced dataset, we divided it into training, validation, and testing sets using our algorithm outlined in Section 3.2 of the main paper. Our proposed method for splitting the data aims to create smaller, non-overlapping regions of concentrated wells by clustering patch centroids. These regions are intended to (a) not intersect, (b) be part of a larger geographic area by clustering initial cluster centroids, and (c) contain a similar distribution of non-well patches. This approach ensures that the train-

ing, validation, and test sets cover all geographic regions, providing a diverse and thorough evaluation. The dataset splits represent various geographical areas, making it a comprehensive benchmark for evaluation and testing. Each dataset split is stored in an HDF5 format file.

- **Are there any errors, sources of noise, or redundancies in the dataset?** If so, please provide a description.
  One limitation of our study is our reliance on well locations provided by the Alberta Energy Regulator, which may not encompass all sites, leading to potential omissions in the ground-truth data. This could result in a lower reported validation and test accuracy, with some correctly predicted well locations being mistakenly categorized as false.

- **Is the dataset self-contained, or does it link to or otherwise rely on external resources (e.g., websites, tweets, other datasets)?** If it links to or relies on external resources, a) are there guarantees that they will exist and remain constant, over time; b) are there official archival versions of the complete dataset (i.e., including the external resources as they existed at the time the dataset was created); c) are there any restrictions (e.g., licenses, fees) associated with any of the external resources that might apply to a dataset consumer? Please provide descriptions of all external resources and any restrictions associated with them, as well as links or other access points, as appropriate.
  The dataset does not rely on the persistence of external resources.

- **Does the dataset contain data that might be considered confidential (e.g., data that is protected by legal privilege or by doctor-patient confidentiality, data that includes the content of individuals' non-public communications)?** If so, please provide a description.
  No.

- **Does the dataset contain data that, if viewed directly, might be offensive, insulting, threatening, or might otherwise cause anxiety?** If so, please describe why.
  No.

### G.3 COLLECTION PROCESS

- **How was the data associated with each instance acquired?** Was the data directly observable (e.g., raw text, movie ratings), reported by subjects (e.g., survey responses), or indirectly inferred/derived from other data (e.g., part-of-speech tags, model-based guesses for age or language)? If the data was indirectly inferred/derived from other data, was the data validated/verified? If so, please describe how.
  The AER publishes AER ST37, a monthly list of wells in Alberta, including location, operation mode, license status, and product type. However, the data needs rigorous quality control as license status, or operation mode may not accurately reflect the actual well status. The authors, with extensive domain expertise, removed duplicate well entries in the metadata and shapefile, keeping the most recent update. We then merge and filter the datasets, categorizing wells as active, abandoned, or suspended based on expert criteria. Duplicate coordinates are resolved by keeping the instance with the latest drill date. We verify all wells are within Alberta's boundaries. After thorough quality control by domain experts, we calculate the geographical bounds covered by wells and divide the region into non-overlapping square patches. These patches include varying numbers of wells, with an equal number of patches with and without wells.

- **What mechanisms or procedures were used to collect the data (e.g., hardware apparatus or sensors, manual human curation, software programs, software APIs)?** How were these mechanisms or procedures validated?
  We acquired multispectral satellite imagery data from Planet Labs, which comprises four bands (RGBN) with a 3-meter-per-pixel resolution obtained through their proprietary API. This data was processed using quality-controlled and cleaned well data to generate segmentation and object detection annotations. The annotations were created using custom Python code, leveraging libraries like Shapely, GeoPandas, and Rasterio, and were validated through visualization using folium and matplotlib.

- **If the dataset is a sample from a larger set, what was the sampling strategy (e.g., deterministic, probabilistic with specific sampling probabilities)?**
  No.

- **Who was involved in the data collection process (e.g., students, crowdworkers, contractors) and how were they compensated (e.g., how much were crowdworkers paid)?** The dataset was a collaborative effort involving the Alberta Energy Regulator, Planet Labs, and the authors. Without the contributions from individuals in these three organizations, this dataset would not have been possible. Proper credit must be given to the authors, Planet Labs, and the Alberta Energy Regulator when using this data.

- **Over what timeframe was the data collected?** Does this timeframe match the creation timeframe of the data associated with the instances (e.g., recent crawl of old news articles)? If not, please describe the timeframe in which the data associated with the instances was created.
  We acquired the data from the Alberta Energy Regulator, specifically from its monthly well bulletin AER ST37 (AER, 2024), dated March 2024. Leveraging domain expertise, we filtered the data to reflect the condition of wells as of September 30, 2023. This decision was made because imagery acquired from Alberta during the winter months tends to have high cloud cover. Therefore, we filtered the data to ensure we could collect the best data for each patch based on satellite data acquired between the summer months of June and September in the region.

- **Were any ethical review processes conducted (e.g., by an institutional review board)?** If so, please provide a description of these review processes, including the outcomes, as well as a link or other access point to any supporting documentation.
  N/A

### G.4 PREPROCESSING/CLEANING/LABELING

- **Was any preprocessing/cleaning/labeling of the data done (e.g., discretization or bucketing,tokenization, part-of-speech tagging, SIFT feature extraction, removal of instances, processing of missing values)?** If so, please provide a description.
  In the Dataset section of our submission, we provide a detailed description of the quality control, cleaning, and labeling processes applied to the data obtained from the Alberta Energy Regulator, which forms the basis of our dataset. The satellite imagery utilized in this project was acquired under the Education & Research license from Planet Labs. We reprojected the raw data to EPSG:32611 using the nearest resampling method. Additionally, we removed all geographic metadata, such as image bounds and coordinates, from the shared data imagery to ensure compliance.

- **Was the "raw" data saved in addition to the preprocessed/cleaned/labeled data (e.g., to support unanticipated future uses)?** If so, please provide a link or other access point to the "raw" data.
  The raw satellite imagery data has been saved for internal use; however, it cannot be shared in its current form. Before sharing, the data must undergo preprocessing to remove metadata, as stipulated by the agreement mentioned earlier.

- **Is the software that was used to preprocess/clean/label the data available?** If so, please provide a link or other access point.
  We plan to share the relevant code used for dataset quality control, patch creation, dataset splitting, data acquisition, and label and HDF5 file creation with the public release of the dataset in the future.

- **Any other comments?**
  N/A

### G.5 USES

- **Has the dataset been used for any tasks already?** If so, please provide a description.
  Currently, there are no public demonstrations of the AWD Dataset in use. In this work, we showcase its application for Binary Segmentation and Binary Object Detection of Well Sites to train algorithms for accurately locating well sites. These algorithms can be scaled across larger regions of interest to compare against existing databases, identifying potentially undocumented wells. Flagging wells not present in databases is crucial, as these could be abandoned wells that are significant emitters of greenhouse gases, making them candidates for plugging.

- **Is there a repository that links to any or all papers or systems that use the dataset?** If so, please provide a link or other access point.
  N/A

- **What (other) tasks could the dataset be used for?**
  Additionally, we provide multi-class labels indicating the operational state of the wells for both cases. These labels can be utilized in future projects for locating wells and classifying their operational status, which will aid in identifying well sites that are not present in government records.

- **Is there anything about the composition of the dataset or the way it was collected and preprocessed/cleaned/labeled that might impact future uses?**
  This dataset focuses on Alberta, Canada, known for its diverse oil reserves and varied landscapes, providing a representative sample comparable to regions in the Appalachian and Mountain West areas of the United States and some former Soviet states with oil wells and unidentified site issues. A limitation of our study is the reliance on well locations from the Alberta Energy Regulator, which may miss some sites, leading to potential false negatives in the ground-truth data. However, this should have minimal impact on algorithm training, as these labels are a minor part of the dataset, and deep learning algorithms can handle moderate label noise well (see e.g., (Rolnick et al., 2017)). The main effect may be underreported test accuracy, with some correctly predicted well locations wrongly counted as false. We plan to investigate this further in future work. Additionally, the use of multi-spectral optical data in the AWD dataset may limit the models' applicability in regions with frequent cloud cover.

- **Are there tasks for which the dataset should not be used?** If so, please provide a description.
  This dataset is intended for non-commercial use only and should not be utilized in any application that could negatively impact biodiversity.

- **Any other comments?**
  N/A

## G.6 DISTRIBUTION

- **Will the dataset be distributed to third parties outside of the entity (e.g., company, institution, organization) on behalf of which the dataset was created?** If so, please provide a description.
  Yes, the dataset will be made public (open-source) in the future.

- **How will the dataset will be distributed (e.g., tarball on website, API, GitHub)?** Does the dataset have a digital object identifier (DOI)?
  The data is currently accessible through a Dropbox folder, which will eventually be migrated to Zenodo. The link to access the data will be provided on our project's GitHub repository.

- **When will the dataset be distributed?**
  The dataset can be downloaded from Dropbox, with the link specified in the main paper and mentioned in the README of the shared codebase for benchmark experiments. Once the submission is made public, the dataset will be hosted on Zenodo, and the link will be provided in the public GitHub repository.

- **Will the dataset be distributed under a copyright or other intellectual property (IP) license, and/or under applicable terms of use (ToU)?** If so, please describe this license and/or ToU, and provide a link or other access point to, or otherwise reproduce, any relevant licensing terms or ToU, as well as any fees associated with these restrictions.
  The AWD Dataset is released under a Creative Commons Attribution-NonCommercial 4.0 International (CC BY-NC 4.0) License (https://creativecommons.org/licenses/by-nc/4.0/).

- **Have any third parties imposed IP-based or other restrictions on the data associated with the instances?** If so, please describe these restrictions and provide a link or other access point to, or otherwise reproduce, any relevant licensing terms, as well as any fees associated with these restrictions.

The satellite imagery used in this project was acquired under the Education & Research license of Planet Labs (PBC, 2024). This license allows for the use of the data in publications and the creation of derivative products, which can be shared in association with publications. However, raw imagery cannot be shared publicly. To comply with these guidelines, we share the data in HDF5 format, with satellite imagery represented as a numpy array from Raster Vector. We have removed all geographic metadata, such as image bounds and coordinates, from the shared data. The data is intended for academic use only and should not be used for commercial purposes. Proper credit must be given to the current authors, Planet Labs, and the Alberta Energy Regulator when using this data.

- **Do any export controls or other regulatory restrictions apply to the dataset or to individual instances?** If so, please describe these restrictions, and provide a link or other access point to, or otherwise reproduce, any supporting documentation.
  No

- **Any other comments?**
  N/A

## G.7 MAINTENANCE

- **Who is supporting/hosting/maintaining the dataset?**
  We are currently hosting the dataset on Dropbox to ensure anonymity. Once it is made public, we plan to host it on Zenodo.

- **How can the owner/curator/manager of the dataset be contacted (e.g., email address)?**
  You can reach the authors through the email addresses provided in the paper once it is made public. Additionally, you can raise any issues on the GitHub repository, which will be made public in the future.

- **Is there an erratum?** If so, please provide a link or other access point.
  Not to the best of our knowledge.

- **Will the dataset be updated (e.g., to correct labeling errors, add new instances, delete instances)?** If so, please describe how often, by whom, and how updates will be communicated to users (e.g., mailing list, GitHub)?
  As our dataset is based on data from a fixed timeframe and consists of satellite imagery collected during a specific period, we do not currently have plans to update it in the near future. However, if there are any changes to these plans, updates to the dataset will be posted on the corresponding GitHub repository once it is made public.

- **Will older versions of the dataset continue to be supported/hosted/maintained?** If so, please describe how. If not, please describe how its obsolescence will be communicated to users.
  If there are newer versions of the dataset, they will maintain the same format. We will ensure that the code associated with the project on GitHub supports these updates, and we will update the READMEs to reflect any changes to the dataset.

- **If others want to extend/augment/build on/contribute to the dataset, is there a mechanism for them to do so?** If so, please provide a description. Will these contributions be validated/verified? If so, please describe how. If not, why not? Is there a process for communicating/distributing these contributions to users? If so, please provide a description.
  We plan to share the relevant code in the future. However, to ensure the ability to compare against our results, we encourage those who wish to build on the dataset to publish their work separately rather than adding to our data repository.

- **Any other comments?**
  N/A

