# OpenReview forum: "Alberta Wells Dataset: Pinpointing Oil and Gas Wells from Satellite Imagery"
_ICLR.cc/2025/Conference — Submitted to ICLR 2025_

### Official Review · Reviewer_Hu4t · 2024-10-20

**Soundness:** 3
**Presentation:** 3
**Contribution:** 4
**Rating:** 6
**Confidence:** 4

**Summary:**

This work presents a large-scale remote sensing multispectral dataset named ALBERTA WELLS, aimed at accurately locating abandoned wells to prevent environmental pollution using remote sensing technology. The dataset, collected using Planet Labs satellites in the Alberta region, encompasses a broad area and a significant number of wells. It provides segmentation and detection annotations using the wells' geographical data and offers a reasonable split into training, validation, and testing sets. The performance of classical architectures on this dataset for segmentation and detection tasks is also provided.

**Strengths:**

1.	The scale of this dataset significantly surpasses other wells datasets, providing ample training data for deep learning. Given the abundance of oil wells in Alberta, creating a dataset for this region is meaningful. The public availability of this valuable dataset greatly facilitates research into the localization of abandoned oil wells.
2.	The manuscript is clearly written; the dataset-splitting method and benchmark experiments are rational.

**Weaknesses:**

The main limitations of this work are the inaccuracies in annotations and the insufficiency of the benchmark experiments, detailed as follows:

1.	There are minor suggestions for improvement in the phrasing. For example, line 431 states, "For the binary segmentation task framing, we train both CNN-based and Transformer-based backbones, considering the prevalent imbalance in the image data due to the small size of wells." This description is somewhat vague. If it implies that the backbones were pre-trained, it should be clearly stated which datasets were used for this pre-training. In Section 4, BENCHMARK EXPERIMENTS, it might not be clear to the reader whether the experiments used only the RGB three channels or included the near-infrared to make four channels. Although Section 3 states that the dataset provides four channels, repeating this in the experimental setup could be beneficial.
2.	Tables 4 and 5 display the benchmark performances of various classical semantic segmentation and object detection architectures on this dataset. While this work does not provide experiments with different model sizes (such as small, base, large versions), this might be sufficient for this dataset. However, including the parameter count and computational load of each benchmark model in the tables would add value to the discussion of segmentation and detection results.
3.	The state-of-the-art backbone, ConvNeXt, was omitted in the experiments. Providing its performance in remote sensing image analysis would make the article's benchmark presentation more comprehensive.
4.	In Figure 4, the satellite images reveal that wells may not always be perfectly circular, yet the semantic segmentation annotations are uniformly circular, and the object detection bounding boxes are uniformly square. This may not accurately reflect the real shape of wells, but the article lacks an explanation for this, except for a mention on line 335 about the "teardrop shape typical for well sites." The accuracy of these annotations might be the primary limitation of this dataset.
5.	Line 336 states that wells "typically range from 70 to 120 meters in diameter," indicating that wells in Alberta are relatively uniform in size. In contrast, other datasets, like the Well Pad Dataset, exhibit more significant size variations, which could impact the performance of few- or zero-shot transfer learning, as mentioned on line 503. Discussing regional differences in well characteristics could enhance this section.

**Questions:**

1.	In the last image of Figure 4, showing multiple wells' bounding boxes, the ground truth for the topmost well overlaps two bounding boxes. What causes this overlap? Is it a shortfall of the quality control strategy introduced in Section 3.1? If so, a detailed discussion in the text would be beneficial.
2.	Line 338 mentions that, although no relevant experiments were conducted, the data annotation also includes the status of wells (active, suspended, or abandoned). I am curious whether these different statuses manifest distinct characteristics in remote sensing images (e.g., in multispectral bands). If the author could provide insights on this, it could enlighten future work.

---

> ### Author Response · Authors · 2024-11-25
>
> Thank you for your careful reading of our paper and the very helpful suggestions. We have attempted to address your comments, including adding significant new experiments regarding the effect of well type and the near-infrared band, as well as the ConvNeXt architecture. Detailed responses are below.
>
> **1. There are minor suggestions for improvement in the phrasing. For example, line 431 states, "For the binary segmentation task framing, we train both CNN-based and Transformer-based backbones, considering the prevalent imbalance in the image data due to the small size of wells." This description is somewhat vague. If it implies that the backbones were pre-trained, it should be clearly stated which datasets were used for this pre-training. In Section 4, BENCHMARK EXPERIMENTS, it might not be clear to the reader whether the experiments used only the RGB three channels or included the near-infrared to make four channels. Although Section 3 states that the dataset provides four channels, repeating this in the experimental setup could be beneficial.**
>
> Thank you for this feedback. Regarding the phrasing in line 431, we appreciate your observation and would like to clarify that the backbones (both CNN-based and Transformer-based) were trained from scratch on our Alberta Wells Dataset, and no pre-training from external datasets was used. Although we did use 3-dimensional, ImageNet initialized weights of the backbone for encoder but modified the initial layers afterwards to support 4-dimensional multispectral images. We have updated the manuscript to explicitly state this.
>
> Regarding the use of RGB vs. RGB+NIR in the benchmark experiments, we confirm that all models were trained using both RGB and Near-Infrared (NIR) channels, not just RGB. We have revised Section 4 (Benchmark Experiments) to make this clear, ensuring that the experimental setup explicitly mentions the use of RGB+NIR data.
>
> **2.Tables 4 and 5 display the benchmark performances of various classical semantic segmentation and object detection architectures on this dataset. While this work does not provide experiments with different model sizes (such as small, base, large versions), this might be sufficient for this dataset. However, including the parameter count and computational load of each benchmark model in the tables would add value to the discussion of segmentation and detection results.**
>
> Thank you for this excellent suggestion. We agree that including the parameter count and computational load for each benchmark model would add value to the discussion of the segmentation and detection results in Tables 4 and 5, and we have added this in the revision to compare the various different models we used, e.g. different sizes of ResNet architectures and Transformer-based models.
>
> **3. The state-of-the-art backbone, ConvNeXt, was omitted in the experiments. Providing its performance in remote sensing image analysis would make the article's benchmark presentation more comprehensive.**
>
> This too is an excellent suggestion. We have included initial results for ConvNeXt in the Appendix of the revision, and are currently running additional experiments.
>
> **4. In Figure 4, the satellite images reveal that wells may not always be perfectly circular, yet the semantic segmentation annotations are uniformly circular, and the object detection bounding boxes are uniformly square. This may not accurately reflect the real shape of wells, but the article lacks an explanation for this, except for a mention on line 335 about the "teardrop shape typical for well sites." The accuracy of these annotations might be the primary limitation of this dataset.**
>
> Thank you for this feedback. We agree that the annotations in Figure 4 (circular segmentation masks and square bounding boxes) may not capture the exact shape of some wells. However, the goal of both the object detection and segmentation tasks is ultimately to pinpoint the locations of wells, not their exact shape - these are weak labels that enable training.
> While this may lead to a slightly increased reported loss, we do not expect this to significantly affect the overall performance of the models. Deep learning algorithms are known to be somewhat robust to moderate amounts of label noise, as shown in previous work (e.g., Rolnick et al., 2017). We will include a clearer explanation of this issue in the revised manuscript.

---

> > ### Author Response · Authors · 2024-11-25
> >
> > **5. Line 336 states that wells "typically range from 70 to 120 meters in diameter," indicating that wells in Alberta are relatively uniform in size. In contrast, other datasets, like the Well Pad Dataset, exhibit more significant size variations, which could impact the performance of few- or zero-shot transfer learning, as mentioned on line 503. Discussing regional differences in well characteristics could enhance this section.**
> >
> > Thank you for your insightful feedback. We agree that discussing regional differences in well characteristics will strengthen the manuscript.
> > Characteristic well pad sizes can vary somewhat, based on factors such as location, land use restrictions, and regulations. Both the Well Pad Dataset and our dataset contain some variation in well pad sizes. In Colorado (one of the two areas represented in the Well Pad Dataset), well pad sizes typically range from 1 to 3 acres according to state regulations, corresponding to circles of diameter 70 to 120 meters, in line with the size of well pads in our dataset.
> > We will revise the manuscript to more thoroughly discuss how regional differences in well sizes might impact model performance and transfer learning, and we will incorporate this point into our future research directions. Thank you again for your helpful suggestion.
> >
> > **6. In the last image of Figure 4, showing multiple wells' bounding boxes, the ground truth for the topmost well overlaps two bounding boxes. What causes this overlap? Is it a shortfall of the quality control strategy introduced in Section 3.1? If so, a detailed discussion in the text would be beneficial.**
> >
> > Thank you for your question. We have addressed this in the revised manuscript. The overlap in the ground truth bounding boxes for the topmost well in Figure 4 is due to the nature of oil and gas well sites, where multiple wells are often located very close to each other, especially in densely developed areas. In such regions, wells and their associated infrastructure (e.g., well pads) can overlap spatially, making it challenging to precisely delineate the boundaries of individual wells. This overlap is not a shortcoming of our quality control strategy, but rather a consequence of annotating wells in areas where they are clustered together.
> >
> > **7.Line 338 mentions that, although no relevant experiments were conducted, the data annotation also includes the status of wells (active, suspended, or abandoned). I am curious whether these different statuses manifest distinct characteristics in remote sensing images (e.g., in multispectral bands). If the author could provide insights on this, it could enlighten future work.**
> >
> > Thank you for this excellent question.
> >
> > Abandoned wells may show signs of disrepair, vegetation regrowth, or environmental changes around the site, which could be highlighted in near-infrared bands sensitive to vegetation and land surface properties. Active wells, by contrast, are often associated with visible infrastructure such as equipment, well pads, and roads, which may stand out in RGB and infrared imagery. Suspended wells could exhibit intermediate features, reflecting reduced activity or partially maintained infrastructure.
> >
> > In the revision, we have conducted initial experiments to investigate this variation, by comparing the impact of training models on data that includes all well types versus only active wells. The preliminary results show that training on a mix of well types is beneficial in performance when evaluated across the whole dataset. We are currently running more detailed experiments on this topic for the camera-ready.

---

> ### Comment · Reviewer_Hu4t · 2024-11-25
>
> Thank you for the detailed and thoughtful responses during the rebuttal process, as well as the additional experiments conducted. Your clarifications and revisions have effectively addressed my concerns and significantly enhanced the quality of the manuscript. Based on your replies and updates, I am pleased to maintain my rating.

---

> > ### Author Response · Authors · 2024-11-25
> >
> > Thank you for your swift response, and for helping us to enhance the quality of the paper! Given that you indicate that we have addressed all the weaknesses you raised, could we request that you consider raising your rating to a standard Accept?

---

### Official Review · Reviewer_hd5c · 2024-10-31

**Soundness:** 3
**Presentation:** 3
**Contribution:** 2
**Rating:** 6
**Confidence:** 4

**Summary:**

The paper presents a dataset for detecting oil and gas wells from satellite imagery in Alberta, Canada. The dataset contains a large volume of PlanetScope (~ 3 m spatial resolution) images (> 90,000) with over 200,000 curated well labels. Using a 2-step clustering approach, the images were split into a training, validation, and test set. The authors report binary segmentation results for several deep learning models including convolutional neural networks and a vision transformer. Additionally, results from object detection models are reported as an alternative approach. Models achieve good performance on both tasks (IoU values > 0.6).

This dataset lists several contributions compared to existing datasets, including its large scale, the use of high-resolution Planet imagery, and the availability of well type (abandoned, suspended, and active) data. However, large scale mainly refers to data volume and not to geographic diversity. Furthermore, the paper does not demonstrate the usefulness of the well type data. These limitations reduce the contribution of the paper compared to existing datasets.

**Strengths:**

- The paper is well-structured and easy to read.

- The dataset employs high-resolution PlanetScope imagery which benefits from a high temporal resolution (even better than that of Sentinel-2) compared to the VHR Google Earth imagery used by most existing well datasets. Furthermore, its spatial resolution is better than that of Sentinel-2.

- The data volume, in terms of number of images and well counts, is significantly bigger than that of existing datasets.

- The dataset includes information about well type (active, suspended, and abandoned), unlike existing datasets which supposedly predominantly focus on active wells.

**Weaknesses:**

- Despite the high data volume, the study area is limited to Alberta, Canada. In comparison, an existing dataset (NEPU-OWOD V3) features sites across China and California, US. In general, I argue that adding more data for a specific region (although I also acknowledge that Alberta is a large region) is less important than including oil and gas wells from a diverse set of geographic locations to test the generalization ability of models. The importance of this (specifically the robustness of models to geographic distribution shifts) is also one of the key points in Rolf et al. (2024) which the paper cites.

- Although the well type data is a key contribution of the paper, the authors did not include any experiments to demonstrate its usefulness. Furthermore, the dataset splitting does not seem to take well types into account. I suggest adding well type distributions across the sets to Figure 1.

- The authors state that existing datasets predominantly contain active wells, which limits their applicability in the context of identifying abandoned or suspended wells (l86), while also admitting that the physical appearance of different well types is very similar (l133). The latter would mean that models trained on only active wells generalize to all well types, which contradicts the former. Please clarify this (ideally with experimental results).

Minor weaknesses:
- I suggest adding a subfigure to Figure 3, showing the point density of the image locations.
- There are also some typos and minor grammar mistakes (eg l46: "onshore wells oil and gas wells").
- I also suggest labeling the spatial resolution of Planet data as "high-resolution" instead of "medium-resolution", a term that is usually reserved for imagery with spatial resolutions of > 10 m (e.g. Landsat).
- The authors might also want to mention the SpaceNet 7 dataset which features Planet imagery and includes a study site with oil wells (L15-0434E-1218N_1736_3318_13), even though the dataset was developed for multi-temporal urban monitoring.

**Questions:**

The authors hypothesize that some features, e.g., ground depressions indicating well sites, may be more detectable in the near-infrared band (l323). Have the authors considered running any ablation experiments comparing RGB vs. RGB+NIR?

---

> ### Author Response · Authors · 2024-11-25
>
> Thank you for your careful reading of our paper and the very helpful suggestions. We have attempted to address your comments, including adding significant new experiments regarding the effect of well type and the near-infrared band. Detailed responses are below.
>
> **1. Despite the high data volume, the study area is limited to Alberta, Canada. In comparison, an existing dataset (NEPU-OWOD V3) features sites across China and California, US. In general, I argue that adding more data for a specific region (although I also acknowledge that Alberta is a large region) is less important than including oil and gas wells from a diverse set of geographic locations to test the generalization ability of models. The importance of this (specifically the robustness of models to geographic distribution shifts) is also one of the key points in Rolf et al. (2024) which the paper cites.**
>
> Thank you for your feedback. We agree completely that the global generalization capability of remote sensing models is an important issue - however, we believe this need not be the only priority for a benchmark dataset. Our results show that well detection in Alberta already presents a meaningful challenge for remote sensing models. Alberta, within itself, encompasses a wide range of landscapes, including prairies, forests, lakes, and mountains. Furthermore, Alberta has the third largest oil reserves in the world, so by itself it represents a key use case for well detection, even if algorithms were to be applied nowhere else.
>
> While NEPU-OWOD V3, as mentioned, does encompass such large-scale geographic diversity, it contains only 722 well images with 3.7k wells. By contrast, our dataset includes over 213k wells. This makes it the first benchmark dataset suitable for training state-of-the-art deep learning models from scratch on the problem of well detection.
>
> **2. Although the well type data is a key contribution of the paper, the authors did not include any experiments to demonstrate its usefulness. Furthermore, the dataset splitting does not seem to take well types into account. I suggest adding well type distributions across the sets to Figure 1.**
>
> Thank you for your feedback. We completely agree that demonstrating the impact of well type data is important  Regarding dataset splitting, we have added plots showing the well type distributions across the dataset splits in Section D4 (Dataset Size & Distribution of Samples) of the appendix.  We did not explicitly take well types into account in our splitting algorithm because the distributions of abandoned and suspended wells are very similar to  those of active wells. Indeed, in the new figure one can see that the frequencies of active, suspended, and abandoned wells are similar across splits.
> Regarding the effect of training on different well types, we have conducted initial experiments comparing the impact of training models on data that includes all well types versus only active wells. The preliminary results show that training on a mix of well types is indeed beneficial in performance when evaluated across the whole dataset. We are currently running more detailed experiments on this topic for the camera-ready.
>
>
> **3. The authors state that existing datasets predominantly contain active wells, which limits their applicability in the context of identifying abandoned or suspended wells (l86), while also admitting that the physical appearance of different well types is very similar (l133). The latter would mean that models trained on only active wells generalize to all well types, which contradicts the former. Please clarify this (ideally with experimental results).**
>
> Thank you for this question. Our explanation wasn’t clear, and we have attempted to improve it and added additional experiments.
> The key point here is that while the physical structure of active, suspended, and abandoned wells can be quite similar, the environmental context surrounding abandoned and suspended wells can differ significantly from active ones, depending on the time at which the well was abandoned. Abandoned wells undergo degradation over time (e.g., vegetation growth, land subsidence, and surface changes), which can make them more challenging to detect, even though their basic infrastructure might appear similar to active wells. Thus, training on suspended and abandoned wells is important if models are to be used in detecting them.
>
> In the revision, we have added preliminary experiments to compare the performance of algorithms in predicting abandoned wells when trained using only active wells versus a mix of active and abandoned wells, which show that the latter is indeed beneficial.
> We hope this explanation clarifies the apparent contradiction, and we plan to include further experimental results in the camera-ready version. Thank you again for your thoughtful feedback.

---

> > ### Author Response · Authors · 2024-11-25
> >
> > **4. I suggest adding a subfigure to Figure 3, showing the point density of the image locations.**
> >
> > Thank you for your suggestion. We agree that adding a subfigure to illustrate the point density of the image locations would provide additional clarity .We have included this subfigure in the appendix of the revision (in section D.4 DATASET SIZE & DISTRIBUTION OF SAMPLES) and will update this in Figure 3 in the camera-ready.
> >
> > **5. There are also some typos and minor grammar mistakes (eg l46: "onshore wells oil and gas wells").**
> >
> > Thank you for pointing out this typo. We have corrected this and will review the manuscript thoroughly to address any other minor grammar mistakes and typos.
> >
> > **6. I also suggest labeling the spatial resolution of Planet data as "high-resolution" instead of "medium-resolution", a term that is usually reserved for imagery with spatial resolutions of > 10 m (e.g. Landsat).**
> >
> > Thank you for this suggestion - we have made the change. In the context of well detection, most available datasets have resolutions of either around 10 meters per pixel or 0.1-0.5 meters per pixel. In the revision, we will note that PlanetScope imagery with a resolution of 3 meters per pixel falls in between these two extremes, while observing that, as you note, it is still a form of high-resolution imagery..
> >
> > **7. The authors might also want to mention the SpaceNet 7 dataset which features Planet imagery and includes a study site with oil wells (L15-0434E-1218N_1736_3318_13), even though the dataset was developed for multi-temporal urban monitoring.**
> >
> > Good point - we have mentioned the SpaceNet 7 dataset in our manuscript to highlight the use of Planet data in related domains.
> >
> > **8. The authors hypothesize that some features, e.g., ground depressions indicating well sites, may be more detectable in the near-infrared band (l323). Have the authors considered running any ablation experiments comparing RGB vs. RGB+NIR?**
> >
> > Thank you for your excellent suggestion. We have run the experiment comparing RGB vs RGB+NIR on both Segmentation (using U-Net with ResNet50 Backbone) and Detection (using FCOS Backbone). In both cases, the addition of the NIR band does provide an additional performance boost of 1-2%. We have included the results in the Appendix of the revision.

---

> > > ### Comment · Reviewer_hd5c · 2024-11-25
> > >
> > > I would like to thank the authors for their comprehensive response. I particularly appreciate the effort to conduct additional experiments. Based on the revisions, I have increased my score, especially since the updated paper demonstrates a use case for the well-type data.
> > >
> > > That said, I would like to point out that some weaknesses of the paper remain. For example, there are several similar datasets, albeit smaller in data volume, already available. Additionally, the geographic coverage of the proposed dataset remains limited. I also share the concerns raised by another reviewer regarding the dataset split and its ability to adequately test the generalization capability of models. Although the dataset includes diverse landscapes, these landscapes appear to be represented across all splits. For instance, in real-world scenarios, the split setup shown in Figure 3.d, where training and validation areas are consistently located near test areas, is less representative and could limit the robustness of the evaluation.
> > >
> > > Finally, I recommend including per-class accuracy values for well types I, II, and III in Table 9. This addition would help clarify whether incorporating other well types enhances overall performance or specifically improves results for those types underrepresented in other datasets.
> > >
> > > A minor stylistic comment: I noticed that the authors are inconsistent with the number of decimal places (e.g., Tables 6 and 7). For example, if numbers are rounded to two decimal places, 87 should be displaced as 87.00.

---

> ### Author Response · Authors · 2024-12-02
>
> Thank you for your thoughtful follow-up. We  very much appreciate your recognition of our additional experiments and revisions.
>
> In response to your comments, we would highlight that our dataset is more than 100 times larger in image count than the nearest comparable datasets such as NEPU-OWOD V3, and contains over 50 times the number of wells. For this reason, we believe that our benchmark makes it uniquely possible to test algorithms that require large datasets.  Additionally, our dataset includes a well-distributed set of non-well images, which is lacking from many existing datasets.
>
> Indeed, we do not claim that a model trained purely on Alberta data would necessarily be able to generalize zero-shot to other regions or landscapes. Rather, since this is a benchmark dataset, the goals are threefold: (i) to provide a testbed for algorithms on the important task of well-detection - Alberta makes it possible to identify algorithms that will be able to train on and perform well across diverse landscapes, but as in other tasks these algorithms would likely still need to be at least fine-tuned within any region of application, (ii) to represent an immediately impactful use case, since detecting wells within Alberta is itself useful, (iii) to provide value to the field of ML with a challenging task.
>
>
>
>
> |   Metric  |   Train Set   | Test Set Active (I) | Test Set Suspended (II) | Test Set Abandoned (III) | Test Set All (I+II+III) |
> |:---------:|:-------------:|:-------------------:|:-----------------------:|:------------------------:|:-----------------------:|
> | Precision |   Active(I)   |      **0.998**      |        **0.997**        |         **0.997**        |        **0.998**        |
> |           | All(I+II+III) |        0.859        |          0.851          |           0.841          |          0.880          |
> |   Recall  |   Active(I)   |        0.502        |          0.501          |           0.500          |          0.502          |
> |           | All(I+II+III) |      **0.576**      |        **0.568**        |         **0.559**        |        **0.624**        |
> |    IOU    |   Active(I)   |        0.502        |          0.500          |           0.500          |          0.502          |
> |           | All(I+II+III) |      **0.532**      |        **0.524**        |         **0.514**        |        **0.569**        |
> |  F1 Score |   Active(I)   |        0.503        |          0.501          |           0.500          |          0.503          |
> |           | All(I+II+III) |      **0.557**      |        **0.547**        |         **0.535**        |        **0.607**        |
>
>
>
>
>
>
> We obtained per-class performance metrics for specific well type in the Binary Segmentation Task (Model Architecture : UNet; Backbone : ResNet50) . As observed, the model's performance aligns closely with the full test set results presented in Appendix A.3. These results highlight the importance of including all well types (active, suspended, and abandoned) in the training dataset to ensure comprehensive detection across diverse scenarios. Training solely on active wells results in significantly poorer performance when applied to individual well types and mixed well-type test sets (IoU ~0.5), indicating limited generalization capability. In contrast, training on all well types enhances the model's ability to handle real-world variability, as evidenced by improved IoU, F1 score, and recall for each well type and the entire test set. The improved recall reflects the model's capability to detect a broader range of wells, which is critical for environmental monitoring, where overlooking even a single abandoned well could lead to unaddressed methane emissions or groundwater contamination. While precision is higher for the model trained exclusively on active wells, this is offset by its low recall, which limits its effectiveness in detecting all wells in an image. Therefore, incorporating all well types into the training process ensures a balanced, reliable performance, enabling accurate detection and classification across a wide range of realistic scenarios.
>
> Thanks for pointing out the minor stylistic inconsistencies regarding decimal places, we have now corrected this point.

---

### Official Review · Reviewer_GgR7 · 2024-11-03

**Soundness:** 2
**Presentation:** 2
**Contribution:** 2
**Rating:** 5
**Confidence:** 4

**Summary:**

This paper contributes a large-scale dataset comprising over 213,000 wells
from Alberta with their status as abandoned, suspended, and active. The ground truth data is collected from the Alberta Energy Regulator (AER), detailing the wells and their geographic locations. However, this data cannot be used directly due to licensing restrictions or the presence of duplicate entries.  To address this issue, the authors applied a data filtering and quality control approach with domain experts. Afterwards, they created a well-distributed dataset representing various geographical regions by applying a clustering algorithm. For satellite data the Planet labs multispectral imagery and also frame the task of identifying wells applying both object detection and segmentation. Ultimately, the authors generated segmentation maps and object detection annotations for all known wells in the images based on the point labels provided in the AER data.

**Strengths:**

1. The work overall is of standard quality, including a potentially impactful dataset for an important application that may be of significant contribution to the ML community and preliminary experiments testing well-established baselines on the dataset.
2. The authors create a well-distributed dataset by the clustering Algorithm, which is very clear and well-written.
3. The identification task involves both object detection and object segmentation. This process produces segmentation maps and object detection labels for all known wells in the image using the print labels from the AER data.
4. The dataset may be of use to policymakers and other stakeholders involved in climate
action.

**Weaknesses:**

***Major***

1. The actual contribution of the work is not clearly established. The ground truth dataset is taken from AER ST37, where the there all geolocations are available. The dataset contains only that data; no new data is contributed here.
2. The model evaluation lacks in-region and out-of-region performance testing, which is a crucial experiment for remote sensing tasks.
3. The figure quality is very poor, e.g., Figures 3 and 4.
4. For object detection, only axis-aligned models are used. The author has not implemented recent models or an oriented bounding box (OBB) approach, such as YOLO-OBB or Faster R-CNN with Rotated Bounding Boxes, which could potentially enhance performance.
5. Paper writing should be improved.


***Minor***

1. The dataset documentation, AWD_Datasheets_for_Dataset.pdf, follows the NeurIPS 2024 format and includes a footnote stating: "Submitted to the 38th Conference on Neural Information Processing Systems (NeurIPS 2024) Track on Datasets and Benchmarks. It violates the rules of the conference.

**Questions:**

1. How can the author address the challenge of applying the model to a new region where ground truth data is unavailable? This is a concern because Table 4 shows low recall in regional performance. When evaluating the trained model on a new region, there is a high risk that many wells may be missed due to potential changes in pixel density. In such cases, how does the author plan to validate the dataset, given that ground truth data is not publicly accessible?
2. What is the unique utility of this dataset, considering that many standard remote sensing datasets, such as DOTA v2, are already available?

---

> ### Author Response · Authors · 2024-11-25
>
> We would like to express our sincere gratitude for your thoughtful review of our submission. We appreciate your feedback and would like to address the concerns you raised. Below is a detailed response to each of the issues raised in your review. We have also added additional experiments in the revision to support our conclusions on the benefit of near-infrared imagery and the advantage of training on data from multiple well types.
>
> **1. The actual contribution of the work is not clearly established. The ground truth dataset is taken from AER ST37, where all geolocations are available. The dataset contains only that data; no new data is contributed here.**
>
> Indeed, as with many benchmark datasets, we did not gather new field data ourselves - instead the value of data-gathering comes from two factors:
> It is highly unlikely that someone from the ICLR community would be able to understand, much less, use raw data from AER ST37. Besides the errors in the database which must be corrected, it is simply not in a form that a machine learning researcher could parse - it is not formatted for code and is encoded using jargon specific to oil well operators. AER also of course does not include satellite imagery - our release of PlanetScope imagery is owing to a specific agreement with Planet Labs - this imagery would not be made publicly available at all were it not for our work.
> While our dataset is based on the AER data, we have performed extensive quality control, curation and enhancement of this data, in collaboration with domain experts.
>
> Previous datasets have been introduced for this problem which were orders of magnitude smaller than ours. This goes to show the effort involved in creating such a benchmark dataset - if it were not hard, these prior works would likely have been significantly more in-depth.
>
>
> **2. The model evaluation lacks in-region and out-of-region performance testing, which is crucial for remote sensing tasks.**
>
> Thank you for this question. We believe that the geography-based splitting approach we adopted is an effective method for evaluating geographical generalization. Our approach ensures that both the training and test sets cover a broad spectrum of Alberta's landscape, balancing the dataset to capture diverse geographical features, including prairies in the east, mountains and forests in the west. This strategy maintains geographical diversity and mimics real-world conditions.
>
> Our results show that well detection in Alberta already presents a meaningful challenge for remote sensing models, irrespective of generalization outside the region. While future work may wish to generalize our dataset to other regions, there is clearly room for improvement on state-of-the-art within Alberta. Furthermore, with the third largest oil reserves in the world, Alberta by itself it represents a key use case for well detection, even if algorithms were to be applied nowhere else.
>
>
> **3. The figure quality is very poor, especially Figures 3 and 4.**
>
> Thank you for your valuable feedback regarding the quality of the figures. We have replaced Figure 3 with an updated version and have enhanced the quality of Figure 4, which was due to a compression artifact in conversion from high-resolution remote-sensing images.
>
> **4. For object detection, only axis-aligned models are used. The authors have not implemented recent models or an oriented bounding box (OBB) approach, such as YOLO-OBB or Faster R-CNN with Rotated Bounding Boxes, which could potentially enhance performance.**
>
> Thank you for your feedback. We agree these approaches could be promising. Since experiments will require significant additional dataset engineering, we have not been able to complete these during the rebuttal period but can look at adding them for the camera-ready version.
>
>
> **5. The dataset documentation follows the NeurIPS 2024 format and includes a footnote stating: "Submitted to the 38th Conference on Neural Information Processing Systems (NeurIPS 2024) Track on Datasets and Benchmarks," which violates the rules of the conference.**
>
> We apologize for the oversight and any confusion it may have caused. The footnote in the supplementary material on Dropbox referencing the code and data was included by mistake, and we have removed it to ensure full compliance with ICLR submission guidelines. The dataset documentation in the supplementary materials on Dropbox will be updated accordingly.
>
> Although the necessary changes, including plans for hosting the dataset and other updates, were made and included in the appendix of the main manuscript at the time of submission, we have ensured that these updates are clearly reflected in the revised version available on Dropbox as well.

---

> > ### Author Response · Authors · 2024-11-25
> >
> > **6. How can the author address the challenge of applying the model to a new region where ground truth data is unavailable? This is a concern because Table 4 shows low recall in regional performance. When evaluating the trained model on a new region, there is a high risk that many wells may be missed due to potential changes in pixel density. In such cases, how does the author plan to validate the dataset, given that ground truth data is not publicly accessible?**
> >
> > Thank you for your thoughtful comment. We understand the concern regarding the potential for low recall when applying the model to new regions, especially given the observations in Table 4. However, we would like to clarify a few points regarding our dataset and model performance, and explain how we have addressed this challenge.
> >
> > - Pixel density: PlanetScope imagery is available worldwide. Therefore it is possible to use the same resolution imagery when applying our algorithms to any other location that may be of interest.
> > - Geography-Based Dataset Splitting: We have already implemented a geography-based dataset splitting approach to explicitly simulate both in-distribution and out-of-distribution scenarios. This method ensures that the model is evaluated on regions it has not seen during training, making it robust to geographical variations. By doing this, we simulate real-world conditions where models must generalize to new, unseen areas.
> > - Recall and Precision Balance: Regarding the low recall observed in our results, it is important to note that this is largely due to the skewed distribution of wells in the dataset. In Alberta, most wells are clustered into regions with only 1 or 2 wells, which results in a highly imbalanced distribution. As a result, the model, while performing well on the small number of high-density areas, tends to predict these dominant regions more frequently, which leads to higher precision but lower recall. This is a common challenge in highly imbalanced datasets and is something which will require specialized architectures to address.
> >
> > **7. What is the unique utility of this dataset, considering that many standard remote sensing datasets, such as DOTA v2, are already available?**
> >
> > We appreciate the reviewer’s concern regarding the utility of our dataset in the context of existing remote sensing datasets like DOTA v2. While datasets like DOTA v2 are useful for general object detection tasks, our dataset offers a unique contribution by focusing specifically on oil and gas wells. Unlike general object detection datasets, our dataset is designed for detecting small, infrastructure-related objects in remote, rural environments, such as oil and gas wells. These wells present unique detection challenges due to their small size and placement in sparsely populated areas - our results show that this is indeed a challenging task that is valuable for evaluating remote sensing algorithms.
> >
> > Our dataset includes over 213,000 wells, encompassing abandoned, suspended, and active wells, making it the first large-scale benchmark dataset on this topic. In contrast, the largest comparable benchmark dataset contains a maximum of 10.5k wells. Moreover, this dataset fills a critical gap by providing a resource that supports climate change mitigation efforts. By identifying oil and gas wells, particularly abandoned and suspended ones, which are a significant source of methane emissions, our dataset can help fill gaps in government data on the locations of these wells. This is invaluable to policymakers and stakeholders focused on reducing environmental damage.

---

> ### Comment · Reviewer_GgR7 · 2024-11-25
>
> Thank you sincerely for providing comprehensive responses. I appreciate that some of my concerns have been addressed. As articulated in the authors' rebuttal, their focus lies within the dataset and benchmarking subdomain of ICLR, which aligns well with the scope of an ICLR subject area dataset and benchmark. I upscale my rating from 3 to 5.

---

> > ### Author Response · Authors · 2024-12-01
> >
> > Thank you for revisiting our submission and adjusting your score to 5; we sincerely appreciate your thoughtful engagement. Are there any additional concerns you may have which we can help address during the remainder of the discussion period?
> >
> > We have attempted to address your previously stated concerns by improving figure quality, removing formatting errors, clarifying the dataset's novelty (extensive curation of AER data, integration with PlanetScope multispectral satellite imagery, and creating the first large-scale benchmark for detecting oil and gas wells), and highlighting its alignment with ICLR’s focus on impactful datasets. We have also added additional experiments in the revision to support our conclusions on the benefit of near-infrared imagery and the advantage of training on data from multiple well types.
> >
> > Thank you again for your contributions to improving this submission.

---

### Official Review · Reviewer_E5au · 2024-11-03

**Soundness:** 3
**Presentation:** 3
**Contribution:** 3
**Rating:** 6
**Confidence:** 4

**Summary:**

To cope with the problem that the huge amount of abandoned  oil and gas wells are unrecorded, the paper presents  the first large-scale remote sensing dataset for pinpointing onshore  oil and gas wells . The paper also gives  object detection and binary segmentation algorithms for evaluation. For better split the dataset, a training/validation/evaluation dataset splitting algorithm is also designed to make the evaluation reasonable. Some benchmark experiments are also provided in detail. Some limitations are analyzed. The paper is well organized.

**Strengths:**

1. The first large scale  Wells Dataset is introduced for oil and gas well detection.
2. The dataset construction process is well described which makes the dataset reliabe. Besides, a training/validation/evaluation dataset splititng algorithm based on clustering principle is presented.
3. Evaluation benchmark algorithms are provided to be baseline for the research in this topic.
4. The limitation of the work is explained in detail.

**Weaknesses:**

1. The area is limited in Alberta, Canada which makes the dataset lose the universality.
2. The resolution scale is only 3 meters resolution because the dataset was acquired from PlanetScope-4-Band imagery.
3. the baseline algorithms don't utilized some physical information about wells.

**Questions:**

1. What's the relation of segmentation and detection for wells analysis?
2. Why is the splitting principle using the geographical location?

---

> ### Author Response · Authors · 2024-11-25
>
> Thank you for your thorough and thoughtful review. We appreciate your recognition of the strengths of our work, including the dataset construction, rigorous benchmarking, and the explanation of limitations. Below, we respond to the concerns and questions you raised to provide additional clarity. We have also added additional experiments in the revision to support our conclusions on the benefit of near-infrared imagery and the advantage of training on data from multiple well types.
>
> **1. The area is limited to Alberta, Canada, which makes the dataset lose universality.**
>
>
> We agree that it is a limitation that our dataset is not even larger. However, previous datasets on this topic are limited to much narrower locations, typically with an average of around 5,000 wells, while our dataset covers the entire province of Alberta (an area larger than the UK and Germany combined), encompassing a diverse range of geographical zones with over 217,000 oil wells, by far the largest of any benchmark dataset released on this problem anywhere in the world.. By training machine learning models on this large-scale dataset, we can enhance their robustness and improve their ability to generalize to similar, less-studied regions with fewer labeled data points. Furthermore, Alberta has the third largest oil reserves in the world, so by itself it represents a very impactful use case, even if the algorithms are applied nowhere else.
>
>
> **2. The resolution scale is limited to 3 meters due to the use of PlanetScope-4-Band imagery.**
>
> While the 3-meter resolution of PlanetScope imagery is a limitation, we believe this resolution strikes a balance between coverage and detail at a relatively low cost, representing  a practical choice for large-scale monitoring.
>
> Additionally, PlanetScope data offers some unique advantages: its satellite constellation is updated daily, allowing for consistent imagery across the dataset. Moreover, PlanetScope provides multispectral imagery (RGB + Near-Infrared), which is valuable for identifying features such as well pads and depressions, which may be difficult to detect using standard RGB images. (We have added experiments in the revision that show the benefits of using the near-infrared channel.)
>
> **3. The baseline algorithms do not utilize some physical information about wells.**
>
> We would like to clarify what is meant by "physical information" in this context. In the current baseline evaluation, we focused on standard deep learning approaches for object detection and segmentation, which primarily rely on image data and do not incorporate additional physical features of the wells (such as well pads, tanks, or surrounding infrastructure). These features, while valuable, are often challenging to extract automatically from imagery without expert input or additional data sources, and can vary considerably between individual wells.
>
> **4. What’s the relation of segmentation and detection for wells analysis?**
>
> We offer both object detection and segmentation framings of the task of pinpointing wells, since it is not clear which approach may be most effective. Ultimately, the goal of both tasks is to understand where wells are, a goal for which object detection is the standard approach. However, the generally stereotyped shape of wells makes it possible also to train a segmentation algorithm on well locations; therefore, we have presented this task framing as well.
>
> **5. Why is the splitting principle using the geographical location?**
>
> We split the training, validation, and evaluation data geographically to reduce the risk of spatial autocorrelation, which could cause performance to be overestimated, as well as to ensure the model can generalize effectively to new, unseen regions, reflecting real-world scenarios where well detection models are applied to areas beyond the training data.
>
> Our approach to dataset splitting ensures that the training and test sets are balanced, geographically diverse, and yet non-overlapping. We developed a custom splitting algorithm that clusters data into non-overlapping groups (k1), which are subsets of larger groups (k2). In this setup, one member from each smaller group (k1) is included in both the test and validation sets, while the rest are part of the training set. This approach helps maintain diversity and reduces the risk of data leakage.

---

> > ### Author Response · Authors · 2024-12-01
> >
> > We would like to thank the reviewer again for the thoughtful feedback on our submission. We have attempted to address all questions and comments above, and would like to ask if there are any remaining concerns.

---

### Meta-Review · Area_Chair_P1N2 · 2024-12-17

**Metareview:**

This paper introduces a large-scale dataset of multi-spectral images of wells in the Alberta region, along with their status as abandoned, suspended, and active. After a preprocessing and quality control by domain experts, a clustering algorithm is applied to this data to create a dataset where the well identification was framed as an object detection and segmentation problem, where a number of deep learning algorithms have been evaluated.

This paper originally received contrasting reviews (ratings) leaned to negative (6, 3, 5, 6), which, after rebuttal, become more positive (6, 5, 6, 6). Overall, the paper is surely valuable from the perspective of dataset creation, useful as a benchmark addressing an important problem.

The AC looked at the reviews and the rebuttal and discussions, and despite the majority of positive reviews does not consider this paper suitable for publication at ICLR conferences. This was also discussed and agreed between AC and SAC.

The creation of the dataset was certainly carried out in a sound way, as well as the data has been checked carefully, and validated using a number of deep learning-based detection and segmentation techniques, but I do not see any relevant contribution for the ICLR/ML community. In other words, other than the application target - again, valuable - AC does not see the added value of proposing this data that does not apparently require specific processing or particular design of new DL algorithms.

In other words, I see better this work published in more vision-related conferences, specific to the addressed application, or venues more related to dataset generation and benchmarking.

**Additional Comments On Reviewer Discussion:**

This paper was also discussed between AC and SAC, and the decision has been made accordingly.

---

### Decision · Program_Chairs · 2025-01-22

Reject